# Technical Note: Lessons from and best practices for the deployment of the Soil Water Isotope Storage System

Rachel E. Havranek [1], Kathryn Snell[1], Sebastian Kopf[1], Brett Davidheiser-Kroll[2], Valerie Morris[3], Bruce Vaughn[3]

Rachel Havranek, Kathryn Snell, Sebastian Kopf, Brett Davidheiser-Kroll, Valerie Morris, Bruce Vaughn

[1]Geological Sciences, University of Colorado Boulder, Boulder, 80303, USA
[2]Thermo Fisher Scientific (Bremen) GmbH, Bremen, Germany
[3]Institute of Arctic and Alpine Research, University of Colorado Boulder, Boulder, 80303, USA

*Correspondence to*: Rachel Havranek (rachel.havranek@colorado.edu)

**Abstract.** Soil water isotope datasets are useful for understanding connections between the hydrosphere, atmosphere, biosphere, and geosphere. However, they have been underproduced because of technical challenges associated with collecting those datasets. Here, we present the results of testing and automation of the Soil Water Isotope Storage System (SWISS). The unique innovation of the SWISS is that we are able to automatically collect water vapor from the critical zone at a regular time interval and then store that water vapor until it can be measured back in a laboratory setting. Through a series of quality assurance and quality control tests, we tested that the SWISS is resistant to both atmospheric intrusion and leaking in both laboratory and field settings. We assessed the accuracy and precision of the SWISS through a series of experiments where water vapor of known composition was introduced into the flasks, stored for 14 days, and then measured. From these experiments, after applying an offset correction to report our values relative to VSMOW, we assess the precision of the SWISS at ±0.9‰ and ±3.7‰ for $\delta^{18}O$ and $\delta^{2}H$, respectively. We deployed three SWISS units to three different field sites to demonstrate that the SWISS stores water vapor reliably enough that we are able to differentiate dynamics both between the sites as well within a single soil column. Overall, we demonstrate that the SWISS retains the stable isotope composition of soil water vapor for long enough to allow researchers to address a wide range of ecohydrologic questions.

## 1 Introduction

Understanding soil water dynamics across a range of environments and soil properties is critical to food and water security (e.g. Mahindawansha et al., 2018; Quade et al., 2019; Rothfuss et al., 2021); understanding biogeochemical cycles, such as the nitrogen and phosphorus cycles (e.g. Hinckley et al., 2014; Harms and Ludwig, 2016); and understanding connections between the hydrosphere, biosphere, geosphere and atmosphere (e.g. Vereeken et al., 2022). One approach that can be used to understand water use and movement in the critical zone is the stable isotope geochemistry of soil water (e.g. Sprenger et al., 2016; Bowen et al., 2019). Variations in the stable isotope ratios of oxygen and hydrogen of soil water ($\delta^{18}O$, $\delta^{2}H$) track physical processes like infiltration, root water uptake and evaporation. In particular, stable water isotopes are useful for disentangling complex mixtures of water from multiple sources (e.g. Dawson and Ehleringer, 1991; Brooks et al., 2010; Soderberg et al., 2012; Good et al., 2015; Bowen et al., 2018; Gomez-Navarro et al., 2019; Sprenger and Allen 2020). Despite the long-recognized utility of measuring soil water isotopes for understanding a range of processes (e.g. Zimmerman

et al., 1966; Peterson & Fry., 1987), soil water isotope datasets have been under-produced as
compared to groundwater and meteoric water isotope datasets (Bowen et al., 2019).
The primary barrier to producing soil water isotope datasets has been the arduous nature
of collecting samples. Historically, there are two primary methods for collecting soil water
samples: either digging a pit and collecting a mass of soil to bring back to the lab for subsequent
water extraction or via lysimeter. The former method disrupts the soil profile each time a sample
is collected, inhibiting the creation of long-term records of soil water isotopes. Lysimeters on the
other hand provide the means to collect multi-year soil water isotope datasets (e.g. Green et al.,
2015; Groh et al., 2018; Hinkley et al., 2014; Stumpp et al., 2012, Zhao et al., 2013), but the
choice of lysimeter can affect the portion of soil water (i.e. mobile vs. bound) that is sampled
(Hinkley et al., 2014; Sprenger et al., 2015) and the soil conditions that are sampleable (i.e.
saturation state). Soil water samples collected from both bulk soil samples and lysimeters often
require manual intervention at the time of sampling.
Building off of innovations in laser-based spectroscopy for stable isotope geochemistry,
the ecohydrology community developed a variety of in situ soil water sampling methods over the
last 15 years that enable the creation of high throughput, high precision analyses of soil water
isotopes (e.g. Wassenaar et al., 2008; Gupta et al. 2009; Rothfuss et al., 2013; Volkmann and
Weiler, 2014; Gaj et al., 2015; Oerter et al., 2016; Beyer et al., 2020; Kübert et al., 2020). These
methods have provided insights into a range of ecohydrologic questions from evaporation and
water use dynamics in managed soils (e.g. Oerter et al., 2017; Quade et al., 2018) to better
understanding where plants and trees source their water (e.g. Beyer et al., 2020). These
innovations have allowed researchers to ask new questions about ecohydrologic dynamics, but
current methods require field deployments of laser-based instruments. Field deployments are
technically possible and have been conducted successfully (e.g. Gaj et al., 2016; Volkmann et al.,
2016; Oerter et al., 2017; Quade et al., 2019; Künhammer et al.,2021; Seeger and Weiler., 2021;
Gessler et al., 2022), but require uninterrupted AC power, adequate shelter, as well as safe and
stable operating environments for best results. These prerequisites are often unavailable at many
field sites, especially in more remote locations and for longer sampling time frames. Given these
logistical constraints, these studies have mostly been done near the institutions performing those
studies. Spatial constraints limit the questions that researchers can ask about soil hydrology in
remote and traditionally understudied landscapes. For example, in the geoscience community
there is significant interest in improving the research community's understanding of how and
when paleoclimate proxies (e.g. stable isotope records from pedogenic carbonate, branched
glycerol dialkyl glycerol tetraethers, etc.) form in soils, because that informs our ability to
accurately interpret records from the geologic past. However, those projects commonly have
environmental constraints like soil type or local climate characteristics that may not be located
near institutions performing those studies. To be able to study a broader range of questions about
ecohydrology, there is a need for a system that is capable of autonomously collecting soil water
vapor for isotopic analysis in remote settings.
In this contribution, we report on the further development and testing of a field
deployable system called the Soil Water Isotope Storage System (SWISS). The SWISS was built
to be paired with ACCURELL PP V8/2HF vapor permeable probes that have been previously
tested for soil water isotope applications (Rothfuss et al., 2013; Oerter et al., 2017). Our system
uses three basic components to store water vapor produced by the vapor permeable probes: glass
flasks, stainless steel tubing and a flask selector valve (Fig. 1, Supplemental Table 1).
Previously, we demonstrated through a series of lab experiments that the glass flasks used in the
SWISS units can reliably store water vapor for up to 30 days (Havranek et al., 2020). That proof-
of-concept study demonstrated that the flasks retain original water isotope values, but the
laboratory system was not field deployable and did not have customizable automation. Here, we
present a fully autonomous, field-ready system that has been tested under both laboratory
conditions and field conditions, including development and testing of a solar-powered, battery
backed automation system that enables pre-scheduled water vapor sampling without manual
intervention in remote field locations.
To test the accuracy and precision of the SWISS, we completed quality assurance and
quality control (QA/QC) tests. Here, we demonstrate the viability of this system under field-
conditions through two field suitability experiments. In addition, we sampled three different field
sites to show that the automation schema works on a monthly timescale and that the system
preserves soil water vapor isotope signals with sufficient precision to distinguish between three
different field settings and vertical profile differences.
2 Field Sites
*2.1 Site Set-Up*
At each site we dug two holes; figure 1 shows the field-setup employed at all of our field
sites.  One hole was instrumented with soil moisture and temperature data loggers at 25 cm, 50
cm, 75 cm, and 100 cm depths, as well as the water vapor permeable probes at 25 cm, 50 cm and
75 cm depths (Fig 1A). We deployed all probes  >9 months before the first samples were
collected to allow the soil to settle and return to natural conditions as much as possible. This
timeframe was longer than other studies (e.g. Kübert et al., 2020) and included infiltration of
spring and early summer precipitation. During probe deployment we took care to retain the
original soil horizon sequence and horizon depths as much as possible. In the second hole, we
stored the SWISS unit, dry nitrogen tank, and associated components to power the SWISS (Fig
1B). The water vapor probes, which connected to the SWISS units with Bev-A-Line
impermeable tubing, were run through a PVC pipe buried at approximately 15 cm depth. We ran
the impermeable tubing underground to limit the effect of diurnal temperature variability on the
impermeable tubing to prevent condensation as water travels from the relatively warm soil to the
SWISS.

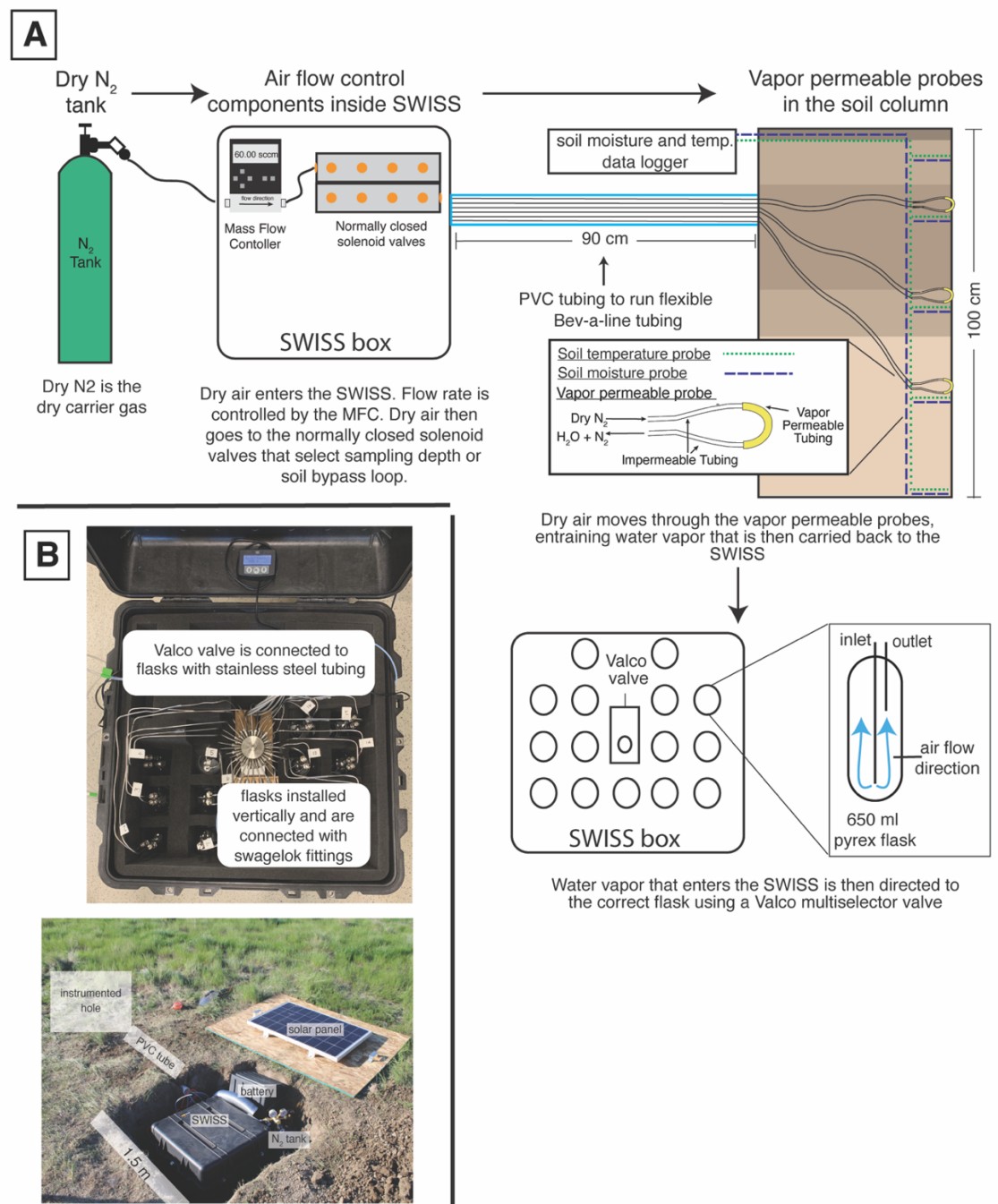

**Figure 1.** A) The sampling flow path. To sample soil water, dry nitrogen is regulated at a specific rate
using a mass flow controller, and then directed to one of the three sampling depths, or the soil bypass loop
using a set of solenoid valves. Both the mass flow controller and solenoid valves are housed inside the
SWISS. Once directed to the correct sampling depth, dry nitrogen is carried to the vapor permeable
probes via gas impermeable tubing that is buried approximately 15 cm depth. After passing through the
vapor permeable probe, the entrained soil water vapor is carried back to the SWISS where it is directed to
the correct flask using a Valco multiselector valve. B) Photos of a built-out SWISS and the layout of a
field site. Each of the system components (solar panel, battery, $N_2$ tank, SWISS, PVC tube) are labeled, in
addition to the location of the instrumented hole in which all of the probes are buried. The hole which
houses the SWISS, power, and $N_2$ tank is approximately 1.5 m wide.
*2.2 Site descriptions*
We deployed the SWISS at three field locations: Oglala National Grassland, Nebraska,
USA; Briggsdale, Colorado, USA; and Seibert, Colorado, USA. The Oglala National Grassland
site (Lat: 42.9600/Long:, -103.5979/elev: 1117 m) is located in northwestern Nebraska, USA in a
cold semi-arid climate. The soil at this site is described as an Aridisol with a silt-loam texture. It
is part of the Olney series (Natural Resources Conservation Service, 2022). The Briggsdale site
(Lat: 40.5947/Long: -104.3190/elev: 1480 m) is located in northeastern Colorado, USA in a cold
semi-arid climate. The soil at this site is described as an Alfisol with a loamy sand - sandy loam
texture. It is part of the Olnest series (Natural Resources Conservation Service, 2022). Long term
meteorological data from the Briggsdale site is available from the co-located CoAgMet site
(CoAgMet, Colorado Climate Center). The Seibert site (Lat: 39.1187/Long: -102.9250/Elev:
1479 m) is located in eastern Colorado, USA in a cold semi-arid climate. The soil at this site has
been described as an Alfisol, that has a sand loam texture in the top 50 cm of the profile, and a
silt loam texture between 50 - 100 cm. It is part of the Stoneham series (Natural Resources
Conservation Service, 2022). Long term meteorological data from the site is available from the
co-located CoAgMet site (CoAgMet, Colorado Climate Center).
**3 Materials**
**3.1 SWISS Hardware components**
In each SWISS there are 15 custom made ~650 ml flasks. These flasks are designed
similarly to those used for other water vapor applications. For example, a similar flask is
currently used in an unmanned aerial vehicle to collect atmospheric water vapor samples for
stable isotope analysis (Rozmiarek et al., 2021). The flasks have one long inlet tube that extends
into the flask almost to the base, and one shorter outlet tube so that vapor exiting the flask is well
mixed and representative of the whole flask (Fig. 1A). The large flask volume is advantageous
because there is a low glass surface area to volume ratio, and therefore we are able to reliably
measure vapor from the flasks on a CRDS instrument without interacting with vapor bound to
the flask walls. The 15 glass flasks are connected to a 16-port, multi-selector Valco valve.  We
chose to use a Valco valve because these have previously been shown to sufficiently seal off
sample volumes for subsequent stable isotope analysis (Theis et al., 2004). The valve and flasks
are connected by 1/8 inch stainless steel tubing and stainless steel 1/4 inch to 1/8 inch union
Swagelok fittings; we use PTFE ferrules on the glass flasks with the Swagelok fittings. The first
port of the Valco valve is 1/8 inch stainless steel tubing that serves as a flask bypass loop, which
enables flushing of either dry air or water vapor through the system without interacting with a
flask. All components are contained in a 61 cm x 61 cm x 61 cm Pelican case (Pelican 0370)
with three layers of Pick n' Pluck foam and convoluted foam (Pelican Products Inc., Torrance,
Ca, USA). This case is thermally insulated and provides enough protection to safely transport the
SWISS by vehicle to field sites.
**3.2 Soil Probes**
There are three components for the collection and analysis of soil water vapor: vapor
permeable probes, soil temperature loggers, and soil moisture sensors (Fig 1B, Supplemental
Table 1).
Here, we use a vapor permeable membrane (Accurrell PP V8/2HF, 3M, Germany) that
was first tested for soil water isotope applications by Rothfuss et al., (2013). This method works
by flushing dry nitrogen (or dry air) through the vapor permeable membrane, creating a water
vapor concentration gradient from inside the probe to the soil, thus inducing water vapor
movement across the membrane. Water vapor is then entrained in the dry nitrogen and flushed to
either a CRDS system or into a storage container. We opted to use this tubing because it has been
shown to deliver reliable data over time (i.e. Rothfuss et al., 2015; Oerter et al., 2019; Kübert et
al., 2020; Seeger and Weiler, 2021; Gessler et al,. 2021), and it is easy to use and customize to
individual needs (Beyer et al., 2020; Kübert et al., 2020). We previously observed that variability
in the length of the vapor permeable tubing can lead to systematic offsets in the stable isotope
composition of measured waters that arise from variability of vapor permeable tube surface area
(Havranek et al., 2020). Therefore, we were careful to construct all probes such that the length
of the Accurrell vapor permeable tubing was 10 cm long, and the impermeable Bev-A-Line IV
connected on each side of the vapor permeable tubing was 2 m long. We cut the Bev-A-Line
connections to identical lengths to control for memory effect and to treat all samples identically.
We also constructed the vapor permeable probes to be used in the lab setting for standards in an
identical fashion.
Soil temperature loggers (Onset HOBO MX2201), used for applying a temperature
correction to all soil water vapor data and to provide key physical parameters of the soils for
other goals beyond this study, were buried at the same depths as the vapor permeable probes.
Soil moisture sensors (Onset S-SMD-M005) were also buried at the same depths as the vapor
permeable probes.
**3. 3 Automation components, code style, and remote setting power**
The philosophy behind the automation of the SWISS was to make it as easy to reproduce
as possible, and as flexible as possible to meet different users' sampling needs. We therefore use
widely available hardware components and electronics parts; for each product there are
numerous alternatives which should be equally viable and could be swapped to better meet each
user's needs. In an effort to make our system as accessible and customizable as possible for the
scientific community, all automation code is completely open source and will continue to be
refined for future applications and hardware improvements. We note that all code is provided as-
is and should be tested carefully for use in other experiments.
The overall sampling scheme used in this paper is described in figure 2 and table 1. Our
experimental goal was to create a time series of soil water vapor data from three discrete
sampling depths (25 cm, 50 cm, 75 cm). Prior to sampling any soil water vapor, we bypassed the
soil probes and flushed the lines within the SWISS. Then, at the start of sampling for each depth,
we also flushed the water vapor probe to remove condensation or 'old' water vapor. The gas
from both of those steps was expelled via the flask bypass loop. Each soil depth was then
sampled for 45 minutes by flushing through the next flask designated in the sequence.
Supplemental figure 1 shows the components of the automation system. To automate and
program the sampling scheme, we used: (1) a microcontroller to run the automation script; (2) a
coin-cell battery powered real time clock so that the microcontroller was always capable of
keeping track of time through power losses, and therefore maintain the sampling schedule; (3) an
RS-232 to TTL converter for serial communication with the Valco valve; (4) solenoid valves that
were used to control which depth was being sampled and the associated direct current (VDC)
power relay; (5) a mass flow controller used to control the rate at which dry nitrogen (1 ppm
$H_2O$) is flushed through the probes; and (6) a power relay used to power the Valco valve and
mass flow controller. All parts are described in detail in Supplemental Table 2.

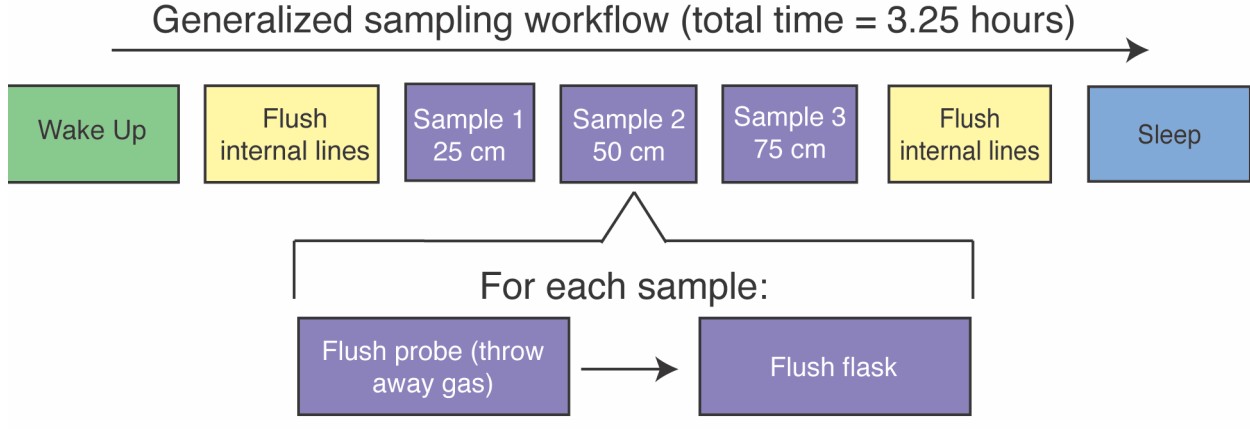

**Figure 2.** Flow chart of the instrument schedule used for sampling during all field experiments.

**Table 1.** Description of soil water sampling steps

| Code Step | Wake-up | Flush internal lines | Flush depth 1 | Sample depth 1 | Flush depth 2 | Sample depth 2 | Flush depth 3 | Sample depth 3 | Flush internal lines | sleep |
|---|---|---|---|---|---|---|---|---|---|---|
| time (minutes) | 1 | 15 | 10 | 45 | 10 | 45 | 10 | 45 | 15 | 1 |
| Valco valve position | flask bypass | flask bypass | flask bypass | 2, 5, 8,11, or 14 | flask bypass | 3, 6, 9, 12, or 15 | flask bypass | 4, 7, 10, 13, or 16 | flask bypass | flask bypass |
| solenoid valve position | none | soil bypass | 25 cm | 25 cm | 50 cm | 50 cm | 75 cm | 75 cm | soil bypass | none |

In a remote setting, the SWISS units are powered using the combination of a 12 volt deep-cycle battery with a 12VDC, 100 watt solar panel that is used to charge the battery. The solar panel is mounted to a piece of plywood that covers the hole where the SWISS is deployed (note, the hole is uncovered in Fig. 1B for illustrative purposes). We opted for this setup because the underground storage of all parts of the system creates a discreet field site that attracts minimal attention from other land users, and helps reduce exposure to temperature and precipitation extremes. In the field, we used a 12VDC-120VAC power inverter to provide simple plug and play power for the Valco valve and mass flow controller. This simple combination was suitable for summertime in the Western U.S. where there are many hours of direct sunlight, and the solar panel was able to easily charge the 12V battery. This setup may need to be adjusted based on location and desired sampling time. Like the automation system, there are many commercial options available for products, and they can be easily adjusted for users' needs; example parts are described in detail in Supplemental Table 2. We also note that in areas where it is possible to plug into a power grid, the deep cycle battery, solar panel and power inverter can be removed.

# 4. Methods

      We completed all water vapor isotope analyses in the Stable Isotope Lab at the Institute of Arctic and Alpine Research (INSTAAR SIL) at the University of Colorado Boulder between October 2020 and August 2022. We used a Picarro L-2130*i* water isotope analyzer (Picarro, Inc. Santa Clara, CA) to measure both water concentration and the oxygen and hydrogen isotope ratios of the water vapor.

## 4.1 QA/QC: Testing the SWISS under lab conditions

      Our highest order concern for the SWISS is that it remains leak-free, because leaks would introduce the potential for fractionation or mixing of atmosphere that would alter the stable isotope ratio of the water vapor in the flask. To mitigate leaks, we developed a three-part quality assurance and quality control (QA/QC) procedure that must be completed for each new SWISS prior to the first deployment. The first step detects any large, fast leaks using helium detection methods; the second step detects medium scale leaks using dry air; and the third step detects slow, small scale leaks using water vapor tests. Full procedural descriptions are available in the supplemental material and the data processing code is available via GitHub.

### 4.1.1 Step 1: Use helium to detect large, fast leaks

      After initial assembly of the SWISS units, we looked for large leaks from the cracking of inlet or outlet tubes on the glass flasks that occasionally occurred while tightening the Swagelok fittings. To do this, we filled the flasks with helium and used a helium leak detector (Leak Detector, Catalog #22655, Restek, Bellefonte, PA, USA). Another easy alternative to a helium leak test is to complete a very short dry air test (methods described below) where the hold-time is on the order of 12-24 hours.

### 4.1.2 Step 2: Use dry air to detect medium scale leaks

      The goal of this test was to catch any second order, medium-scale leaks associated with either Valco valve fittings or Swagelok fittings that were under tightened.

*Step 2A: Fill flasks with dry air*

      To start every experiment, we filled flasks with air that is filtered through Drierite (which has a water vapor mole fraction of less than 500 ppm), at 2 L/min for 5 minutes. With a flask volume of 650 ml, this means the volume of the flask is turned over 15 times.

*Step 2B: Hold period*

      Flasks were then sealed and left to sit for seven days. This time period can be adjusted by other users to fit their climate or needs.

*Step 2C: Measure water vapor mole fraction using dead-end pull sample introduction*

      At the end of the seven-day period, we measured each flask using a dead-end pull sample introduction method. For this sample introduction method, the inlet to the Valco valve was sealed with a 1/4 inch Swagelok cap and there was no introduction of a carrier gas. As a result, air was removed from the flask based on the flow rate of the Picarro analyzer (typically 27 - 31 ml/min). Flasks were measured for five minutes, which resulted in ~150 ml of air being removed from the flasks. All components within the SWISS are capable of being fully evacuated. Water vapor mole fractions determined by Picarros are not standardized, so it is impossible to know for

sure the exact magnitude of water vapor mole fraction change between the input analysis and the
final value at the end of the dry air test. However, these instruments are remarkably stable over
weeks, and so the relative changes observed (e.g. increase or decrease of mole fraction relative to
the initial amount) are likely reliable, particularly for the larger magnitude changes.
If a flask had a water vapor mole fraction of less than 500 ppm, it "passed" step 2 of
QA/QC. If a flask had a water vapor mole fraction greater than 500 ppm, it "failed" step 2 of
QA/QC, and we tightened both the Swagelok connections on the flasks as well as the fittings
between the stainless steel tubing and the Valco valve. We repeated dry air tests on any given
SWISS unit until the majority (typically at least 13/15) of the flasks had passed step 2 of QA/QC.
### 4.1.3 Step 3: Water vapor tests detect small scale leaks
The purpose of this experiment was to mimic storage of water vapor at concentrations
similar to what we might expect in a soil, and for durations similar to those of our field
experiments. These experiments were meant to test whether flasks filled early in the sampling
sequence during field deployments leak by the time samples are returned to the lab for
measurement. For this experiment, we filled flasks with water vapor of known isotopic
composition and water vapor mole fraction, sealed the flasks for 14 days, and then measured the
water vapor mole fraction and isotope values of each flask. We performed 11 water vapor tests
that were done across three analytical sessions using six different SWISS units. Across these
three sessions, we measured 164 flasks both at the start of the 14-day experiment, and at the end.
*Step 3A: Flush flasks with dry air*
Prior to putting any water vapor into the flasks (either in the field or in the lab), we
completed a dry air fill (as described in QA/QC step 2A) that served to purge the flasks of any
prior water vapor that might exchange with the new sample.
*Step 3B: Fill flasks with water vapor and measure input isotope values*
To supply water vapor to the flasks, we used the vapor permeable probes that were
constructed identically to those deployed in the field. We immersed the probes up to the
connection between the vapor permeable and impermeable tubing in water, taking care to not
submerge the connection point and inadvertently allow liquid water to enter the inside of the
vapor permeable tubing. We flushed the flasks at a rate of 150 ml/min for 30 minutes, and
measured the $\delta^{18}O$ and $\delta^{2}H$ values and mole fraction of water vapor as each flask was filled. To
fill 15 flasks sequentially, the probes were submerged in water for approximately 7.5 hours.
Across three different sessions, we used three different waters that are tertiary standards
in the INSTAAR SIL to complete these experiments: a light water made from melting and
filtering Rocky Mountain snow ($\sim$ -25.5‰ and -187.5‰ VSMOW, for $\delta^{18}O$ and $\delta^{2}H$,
respectively), an intermediate water that is deionized (DI) water from the University of Colorado
Boulder Campus ($\sim$ -16.2‰ and -120.7‰ VSMOW for $\delta^{18}O$ and $\delta^{2}H$, respectively) and a heavy
water that is filtered water sourced from Florida, USA ($\sim$ -0.8‰ and -2.8‰ VSMOW for $\delta^{18}O$
and $\delta^{2}H$, respectively). All tertiary lab standards are characterized relative to international
primary standards obtained from the International Atomic Energy Agency and are reported
relative to the V-SMOW/SLAP standard isotope scale. To calculate the input value, we averaged
$\delta^{18}O$ and $\delta^{2}H$ values over the last three minutes of the filling period. We then stored the water
vapor in the flasks for 14 days. At the end of the 14-day storage period, we measured each flask
to evaluate if the $\delta^{18}O$ and $\delta^{2}H$ values had significantly changed over the storage period.
*Step 3C: Measure the water vapor isotope values*
To mitigate memory effects between flasks, we ran dry air via the flask bypass loop (port
one of every SWISS unit) for five minutes between each flask measurement. To verify that the
impermeable tubing between the SWISS and the Picarro was sufficiently dried, we waited until
the water vapor mixing ratio being measured by the Picarro was below 500 ppm for >30 seconds.
During this five-minute window, we used a heat gun to manually warm each flask. We
believe heating the flasks creates a more stable measurement by limiting water vapor bound to
the glass walls of the flask and by helping to homogenize the water vapor within the flask. While
we did not strictly control or regulate the temperature of the flasks, they were all warm to the
touch.
Once we warmed the flask and dried the impermeable tubing, water vapor was introduced
to the CRDS using one of two methods: 1) the dead-end pull sample introduction method
described above, or 2) a *dry air carrier gas sample introduction* method. During the dry air
carrier gas sample introduction method, dry air is continuously flowing through the flask at a rate
of 27-31 ml/min for the entire 12-minute measurement period. To reach a water vapor mole
fraction of approximately 25,000 ppm (the optimal humidity range for the Picarro L2130-*i*), we
diluted the water vapor with dry air at a rate of 10 ml/min. Without dilution, the concentration
out of the flasks is as high as 35,000 - 40,000 ppm, which leads to linearity effects on a Picarro
L2130-*i* that can be challenging to correct for. The dead-end pull method is preferable when the
water vapor mole fraction inside the flask is low (<17,000 ppm), because there is no additional
introduction of dry air. The introduction of dry air decreases the water vapor mole fraction
throughout the measurement, and in fairly dry flasks, using the dry air carrier gas method can
lower the water vapor mole fraction to below 10,000 ppm. Below 10,000 ppm, there are large
linearity isotope effects associated with the measurement on a Picarro L2130-*i*, and the isotope
values are challenging to correct into a known reference frame, just as with high water vapor
mole fractions. The major downside of the dead-end pull method is that condensation is more
likely to form in the stainless steel tubing that connects the flasks to the Valco valve, as well as
the Valco valve itself, compared to the dry air carrier gas method. The dry air carrier gas method
prevents condensation from forming in the Valco valve and tubing, and prevents fractionation
that may occur because of changing pressure within the flask. It is possible that during a dead-
end pull on the flask, heavier isotopes may remain attached to the walls of the flask, coming off
later as the pressure drops. For these reasons, the dry air carrier gas sample introduction method
is our preferred method for sample introduction in most cases.
For each flask, we looked at the stability of the isotope values as well as either a stable
water vapor mole fraction if the dead end pull method was being used or a steady, linear decrease
in water vapor mole fraction if the dry air carrier gas method was being used. For approximately
90% of the flasks we found that after excluding the first three minutes of measurement of each
flask, the subsequent three minutes were the most stable. For the remaining ~10% of the flasks,
using a time window that started either ~30 seconds earlier or ~30 seconds later to create an
average isotope value offered a more stable isotope signal with smaller instrumental
uncertainties. Any flask that required specialized treatment during the data reduction process was
flagged during measurement.
*Step 3D:  Data correction*

During these experiments, we monitored instrument performance (e.g. drift) in two ways.
First, to run standards identically to how samples were collected, we introduced tertiary
standards, described above, using vapor probes. The water vapor produced by the vapor
permeable probes was flushed through the SWISS unit via the flask bypass loop and diluted with
a 10 ml/min dry air flow to reach a water vapor mole fraction of approximately 25,000 ppm
before entering the Picarro. Second, we introduced a suite of four secondary standards that have
been calibrated against primary standards, and reported against VSMOW/SLAP via a flash
evaporator system described in detail by Rozmiarek and others (2021). This flash evaporator
system can be used to adjust the water vapor mole fraction to create linearity corrections at high
and low water vapor mole fractions. After correcting data into a common reference frame, we
calculated the difference between the input isotope values and the ending isotope values.
The results of these tests were used to carefully document flasks that do not perform well,
and any idiosyncrasies of SWISS units. That way, during field deployment suspicious those
flasks could be easily identified and investigated.
**4.2 Field suitability experiments:**
*4.2.1 Field suitability experiment #1: Long term field dry air test*
As a complement to the QA/QC we did under lab conditions, we also completed long
term dry air tests at our field sites. We had three goals associated with these experiments. The
first was to test whether, even under field conditions, where daily temperature and relative
humidity fluctuations are different than in a lab setting, the flasks were still resistant to
atmospheric intrusion. Second, we used these tests to evaluate whether the flasks that were
flushed with soil water vapor near the end of a sampling sequence took on atmosphere prior to
sampling. Lastly, we chose these time intervals because they bracket the typical length of a
deployment, which helped us determine how quickly flasks should be measured after bringing a
SWISS back to the lab.
Like all field deployments, we started with a dry air fill, and then one SWISS unit was
deployed to each of our three field sites. No soil water was collected during these deployments.
The duration between filling the flasks with dry air to measuring the flasks was between 34 - 52
days. The 34 and 52 day tests were done during June 2022 and August 2021, respectively, and
therefore tests the SWISS under warm summertime conditions. The 43 day test was done in
October 2021, which included nights where air temperatures fell below 0°C. The only barrier
between air and the SWISS in its deployment hole was a plywood board, and so this deployment
tested the suitability of the SWISS to maintain integrity under freezing conditions.
*4.2.2. Field suitability experiment #2: Mock field tests*
To test whether the automation code and sampling scheme we developed worked as
expected on short, observable timescales, we set up an experiment to simulate field deployment
of one SWISS unit (Meringue) near the University of Colorado Boulder. This test applied the
automation components and remote power setup described in the materials section. During this
field-simulation experiment, our goal was to collect three discrete samples each sampling period,
to simulate the collection of water vapor from three soil depths. An important goal of this test
was to test whether the sampling scheme introduced any memory effects between samples. We
followed the sampling protocol described in figure 2 and table 1.
The day before the experiment began, all flasks were flushed with dry air as described in
section 4.1.2. Over the course of 25 hours, all 15 flasks were filled with three different vapors
according to a set schedule as would be done in the field. Two of the vapors were created by
immersing the water vapor permeable probes in the
light water and intermediate water as described in section 4.1.3. The third was water vapor from
the ambient atmosphere. All three vapors were sampled using vapor permeable probes
constructed identically to those deployed in the field. For this experiment, we filled three flasks
per cycle with each one of the waters (e.g. Flask 2 = light, Flask 3 = intermediate, Flask 4 =
atmosphere). The choice to sample atmosphere alongside two waters reflects our second goal of
this test, which was to demonstrate that sampled water vapor isotope values do not drift towards
atmospheric values (Magh et al., 2022).
Following the sampling schedule, we stored the SWISS unit in a simulated field setting
for seven days. At the end of the seven days, we measured the flasks. For flasks that had a high
water vapor mole fraction (i.e. light and intermediate water vapor samples) we used the dry air
carrier gas sample introduction method. For flasks that had a low water vapor mole fraction (i.e.
atmosphere, ~15,000 ppm) we used the dead end pull sample introduction method.
To create average values for each flask, we followed the same averaging protocol
described in section 4.1.3. We used equations 2A and 2B from Rothfuss et al., (2013) to convert
from water vapor to liquid values. Then, using secondary and tertiary standards, data were
corrected into the VSMOW isotope scale. Finally, the SWISS unit offset correction (detailed
below in section 6.1.2) was applied.
*4.3 Example Field Deployment: One month period*
We deployed one SWISS unit each to the three field sites described in summer 2022.
Before deployment, all SWISS units were flushed with dry air following the protocol outlined in
section 4.1.2. Flasks were flushed with dry air one to three days prior to field deployment. At
each site, we sampled at three depths (25 cm, 50 cm, and 75cm) on each sampling day, following
the protocol described in figure 2 and table 1. We sampled soil water from all three depths every
five days (protocol length = 25 days total). At Oglala National Grassland, samples were taken
every five days from 2022-06-25 to 2022-07-14.  At the Briggsdale, CO site samples were taken
every five days between 2022-07-17 and 2022-08-06. At the Seibert, CO site, samples were
collected every five days between 2022-06-19 and 2022-07-04. At the end of a 28-day period,
the SWISS units were returned to the lab, and measured. SWISS units were measured within five
days of returning from the field. The maximum number of days a flask held sample water vapor
during these deployments was 32 days. The measurement protocol and data averaging protocol
follows the procedures described in section 4.1.3. The data correction scheme follows as in the
section 4.2.2.
# 5 Results
**5.1 QA/QC Results**
*5.1.1 Dry air test*
Figure 3 shows the results of a seven-day dry air test for three SWISS units (marked by
the unit name) (SI Table 3). For all three SWISS units, at least 13/15 of the flasks maintained a
water vapor mole fraction value of less than 500 ppm over the seven-day period. In two of the
three SWISS units (Lindt and Raclette), the water vapor mole fraction for flasks was randomly
distributed around approximately 350 ppm. In Toblerone there was a systematic decrease in
water vapor mole fraction from flask two through flask 16, matching the order in which the
flasks were filled with dry air initially. In all three SWISS units, flask two had the highest water
vapor mole fraction of all the flasks. Supplemental figure 2 shows the results of successive dry
air tests on the SWISS unit Toblerone where Swagelok fittings were tightened between tests.
There was a significant decrease in measured water vapor mole fraction for many flasks, but
particularly for flasks 10 and 11 as a result of tightening the fittings.

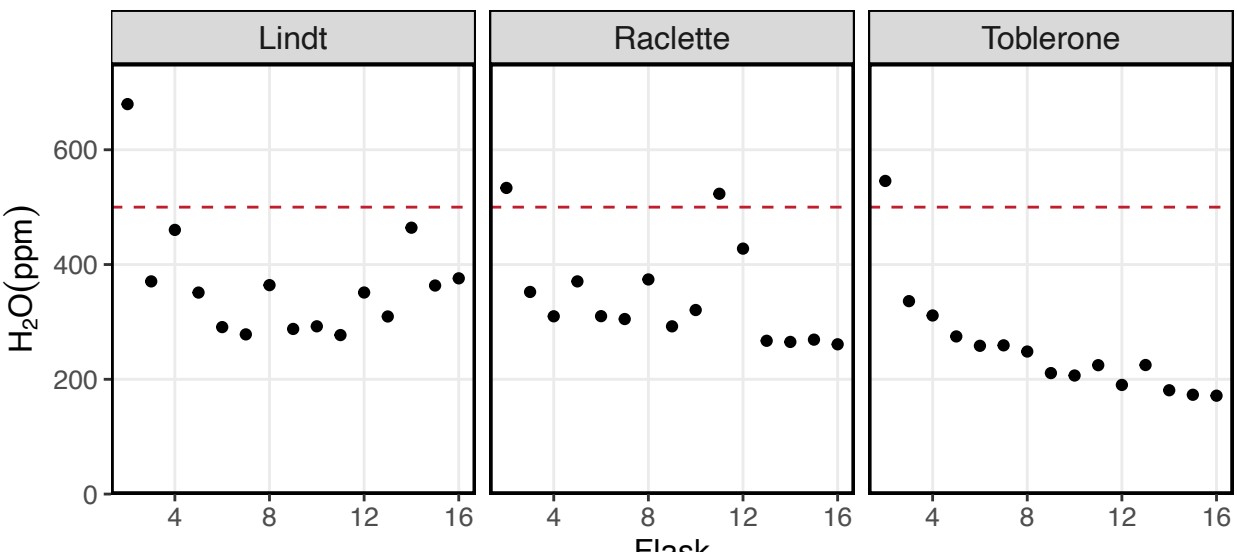

**Figure 3. Results of a dry air test from three different SWISS units named: Lindt, Raclette and Toblerone. The majority**
**of the flasks maintain a water vapor mixing ratio of less than 500 ppm.**

### 5.1.2. Water vapor test
Figure 4 shows the $\delta^{18}O$ results of 11 water vapor tests performed using six different
SWISS units. Ideally, we expect a normal distribution centered about 0 within the uncertainty
limits of the water vapor probes (Oerter et al., 2016). For $\delta^{18}O$, the mean difference between the
start and end values for the flasks is 1.1‰ with a standard deviation of 0.72‰ (outliers
removed). There is a consistent positive offset, with a few clear outliers (Fig. 4A). We do not
observe a consistent difference between water vapor sample introduction methods (Supplemental
Fig. 3). After removing outliers (< Q1 - 1.5*IQR or > Q3 + 1.5*IQR, n = 15) from the dataset,
we compared the kernel density estimate shape to a normal distribution calculated from the mean
and standard deviation of the dataset to assess dataset normality (Fig. 4B). A normal distribution
slightly overestimates the center of the data, but captures the overall shape fairly well. Therefore,
we used the median offset (1.0‰) to correct our water vapor isotope values, and used the
interquartile range of the dataset (outliers removed) to estimate uncertainty of the SWISS as ±
0.9‰ . In figure 5C, for simplicity, we just present the results from 45 flasks (three SWISS
units), with the 1.0‰ offset correction applied. After correction, data are randomly distributed
about 0, and are within the uncertainty range of ± 0.9‰ (Supp. Table 4).
Figure 5 shows the $\delta^{2}H$ results of 11 water vapor tests. For $\delta^{2}H$, the mean difference
between the start and end values is 2.63‰ with a standard deviation of 2.85‰ (outliers
removed). Similar to $\delta^{18}O$, we expected a normal distribution of differences centered around 0.
As with $\delta^{18}O$, there was a consistent positive offset with some outliers (i.e., < Q1 - 1.5*IQR or >
Q3 + 1.5*IQR) (Fig. 5A). After removing outliers (n = 26) from the dataset, we compared the
kernel density estimate to a normal distribution calculated from the mean and standard deviation
of the dataset to assess dataset normality (Fig. 5B). As with $\delta^{18}O$, the center of the dataset is
overestimated by the mean, but the overall peak shape is roughly captured. We therefore use the
median value of 2.3‰ as an offset correction and estimate uncertainty at ±3.7‰ for $\delta^2H$ from the
interquartile range. In figure 5C, we present the results from 45 flasks (three SWISS units), with
the 2.3‰ offset correction applied. Data are randomly distributed about 0 and are within the
uncertainty range of ± 3.7‰ (Supplemental Table 4).
When we compared the results in figures 4C and 5C, we found that flasks that performed
adequately for $\delta^{18}O$ did not always perform adequately for $\delta^2H$. The results from the SWISS unit
Lindt display this behavior particularly well. Less commonly, some flasks that were within
uncertainty of the system for $\delta^2H$ were not within uncertainty of the system for $\delta^{18}O$, like flask 8
in the SWISS unit Toblerone (Figs. 4C, 5C). In a dual isotope plot, there is a strong positive
correlation between $\delta^2H$ and $\delta^{18}O$ with a slope of 3.14 and an $R^2$ value of 0.62 (Supplemental
Fig. 4).


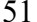

**Figure 4.** $\delta^{18}O$ results of the water vapor tests. A) Boxplot of the difference between the starting $\delta^{18}O$ value and the final $\delta^{18}O$ value of all 164 flasks. B) After removing the outliers from the dataset, the kernel density estimate (black line) and the normal distribution calculated from the dataset (dashed green) are shown. C) After applying the offset correction of 1.0‰, the difference between the starting $\delta^{18}O$ value and the final $\delta^{18}O$ value for three boxes from the August 2022 session are shown. An uncertainty of $\pm$ 0.9‰ is marked with a dashed line, and data points that fall outside that uncertainty are colored red.


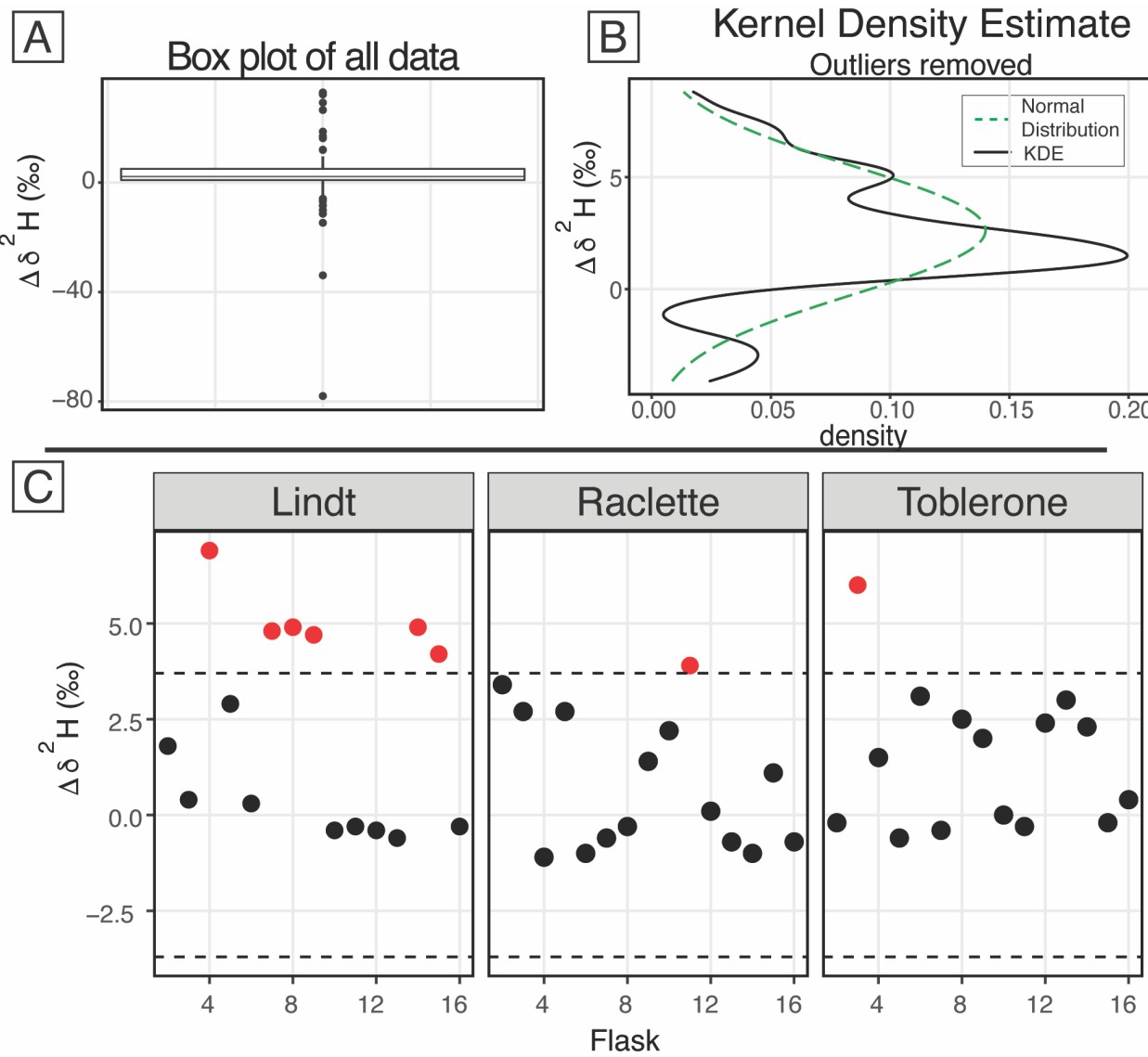

**Figure 5.** $\delta^2$H results of the water vapor tests A) Boxplot of the difference between the starting $\delta^2$H value and the final $\delta^2$H value of all 164 flasks. B) After removing the outliers from the dataset, the kernel density estimate (black line) and the normal distribution calculated from the dataset (dashed green) are shown. C) The difference between the starting $\delta^2$H value and the final $\delta^2$H value for three boxes from the August 2022 session are shown after applying the offset correction of 2.3‰. An uncertainty of ± 3.7‰ is marked with a dashed line, and data points that fall outside that uncertainty are colored red.

## 5.2 Field suitability test results
### 5.2.1 Dry air test
Figure 6A shows the result of placing three different SWISS units that were flushed with dry air out into the field for 34 - 52 days (SI Table 3). This timescale (4-6 weeks) is similar to most field deployments. At the timescale of 34 - 43 days, 13 of the 15 flasks typically maintained a water vapor mole fraction of less than 1000 ppm.  Over the 52 days, seven flasks maintained a water vapor mole fraction less than 1000 ppm and the remaining 8 had a water vapor mole fraction between 1000 - 2500 ppm.

## 5.2.2 Automation test
Figure 6B shows the results of using the automation code to collect and store water vapor of known composition for seven days (Table 2). In both plots, the known values of the water are shown as a long-dash line. Uncertainty on those measurements is estimated at ±0.5‰ and ±2.4‰ for $\delta^{18}O$ and $\delta^{2}H$, respectively (Oerter et al., 2016), shown as the dotted lines. We estimated the isotope value of the atmosphere at the time of sampling with water vapor mole fraction, $\delta^{18}O$, and $\delta^{2}H$ data from the CRDS in the lab. The isotope value, that was corrected as described in section 4.2.2, of each flask is shown, with uncertainty associated with the SWISS units estimated at ±0.9‰ and ±3.7‰ for $\delta^{18}O$ and $\delta^{2}H$, respectively.

Seven of the nine flasks filled with flash-evaporated water vapor overlap within uncertainty of the known $\delta^{18}O$ value for those standards (top plot, Fig. 6B), and four of the five flasks filled with atmospheric vapor overlap within uncertainty of our estimated $\delta^{18}O$ value. Flasks that fall outside of the bounds of uncertainty have lower $\delta^{18}O$ values than the expected value. For $\delta^{2}H$, (bottom plot, Fig. 6B) only three of the nine flasks filled with flash-evaporated water vapor overlap within uncertainty of the known value of those standards, while four of the five flasks filled with atmospheric vapor overlap within uncertainty of the estimated $\delta^{2}H$ value. Flasks that fall outside of the bounds of uncertainty have higher $\delta^{2}H$ values than the expected value.

**Table 2.** Results of the Automation test

| SWISS | Flask | water | $\delta^{18}O$ (‰) | $\delta^2H$ (‰) |
|---|---|---|---|---|
| Meringue | 2 | DI | -14.4 | -122.2 |
| Meringue | 3 | Atmosphere | -10.1 | -105.6 |
| Meringue | 4 | Light | -24.6 | -193.7 |
| Meringue | 5 | DI | -15.0 | -130.8 |
| Meringue | 6 | Atmosphere | -9.4 | -103.4 |
| Meringue | 7 | Light | -25.1 | -201.5 |
| Meringue | 8 | DI | -17.3 | -140.5 |
| Meringue | 9 | Atmosphere | -9.1 | -98.4 |
| Meringue | 10 | Light | -23.7 | -200.7 |
| Meringue | 11 | DI | -14.1 | -122.5 |
| Meringue | 12 | Atmosphere | -8.7 | -94.5 |
| Meringue | 13 | Light | -22.7 | -181.2 |
| Meringue | 14 | DI | -15.2 | -120.5 |
| Meringue | 15 | Atmosphere | -9.2 | -101.1 |
| Meringue | 16 | Light | -23.3 | -192.9 |


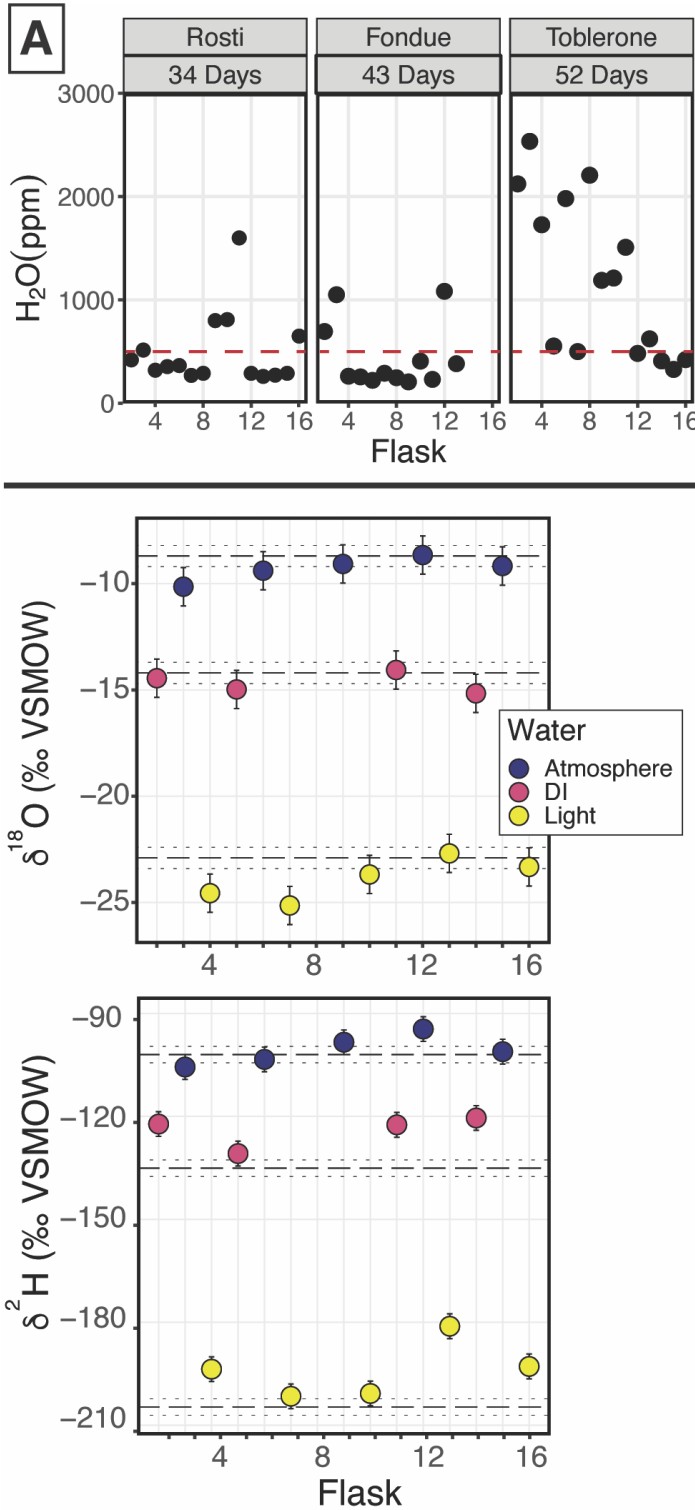

**Figure 6.** A) Results from three different field-based long dry air tests. B) Results from the automation
field suitability tests using the SWISS unit named Meringue. Flasks that sampled atmosphere are shown
in blue, flasks that sampled deionized water (DI) are shown in pink, and flasks that sampled the light
water are shown in yellow. The top plot shows the $\delta^{18}O$ results, and the bottom plot shows the $\delta^2H$ results.

*5.3 Example Field deployment results*
Figure 7 shows the results from three field deployments in Oglala National Grassland,
Nebraska; Briggsdale, Colorado; and Seibert, Colorado (table 3).

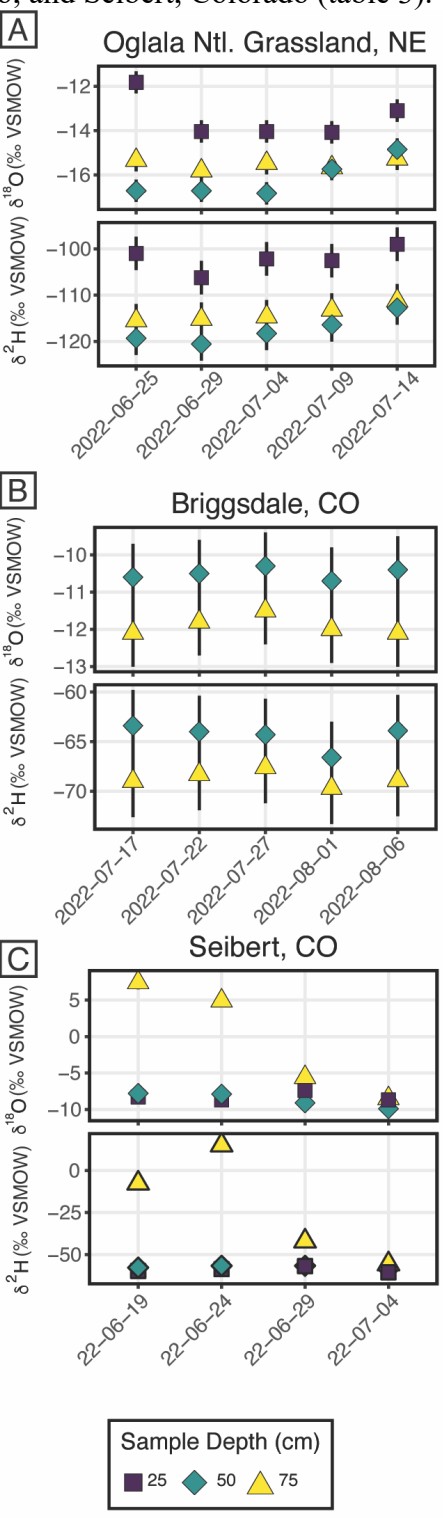

**Figure 7.** Results from all three field deployments to A) Oglala National Grassland, NE, B) Briggsdale,
CO and C) Seibert, CO. Note, the y-axis scale for all three plots is different.

There are 15 samples from Oglala National Grassland (Fig. 7A, Table 3); five from 25
cm depth, five from 50 cm depth and five from 75 cm depth. Four of the five samples from 25
cm overlap within uncertainty in $\delta^{18}O$ value, and all five samples overlap with uncertainty in $\delta^2H$
value. There is a significant decrease in the $\delta^{18}O$ value at 25 cm between 2022-06-25 and 2022-
06-29. There is no similar shift in $\delta^2H$ value over the same time period. The first three samples
from 50 cm overlap in both $\delta^{18}O$ and $\delta^2H$ values, then the final two samples shift to higher
isotope values. Similar to the samples from 50 cm, there is a trend towards higher $\delta^2H$ values for
the last three samples. All five samples from 75 cm overlap in $\delta^{18}O$ and $\delta^2H$ values. On a dual
isotope plot, data from 50 cm and 75 cm cluster together at lower values, while the $\delta^{18}O$ and $\delta^2H$
values from 25 cm are higher (Figs. 7A, 8A). All of the data overlap within uncertainty with the
global meteoric water line, except for the 25 cm depth sample from 2022-06-25 (Fig. 8A). The
calculated D-excess values are all within uncertainty of 10‰ and each other between 2022-06-29
and 2022-07-14 (Fig 8B), except for the 25 cm depth sample from 2022-06-25, which has a D-
excess value of -6.6‰, consistent with evaporative enrichment of soil water at that depth and
time.

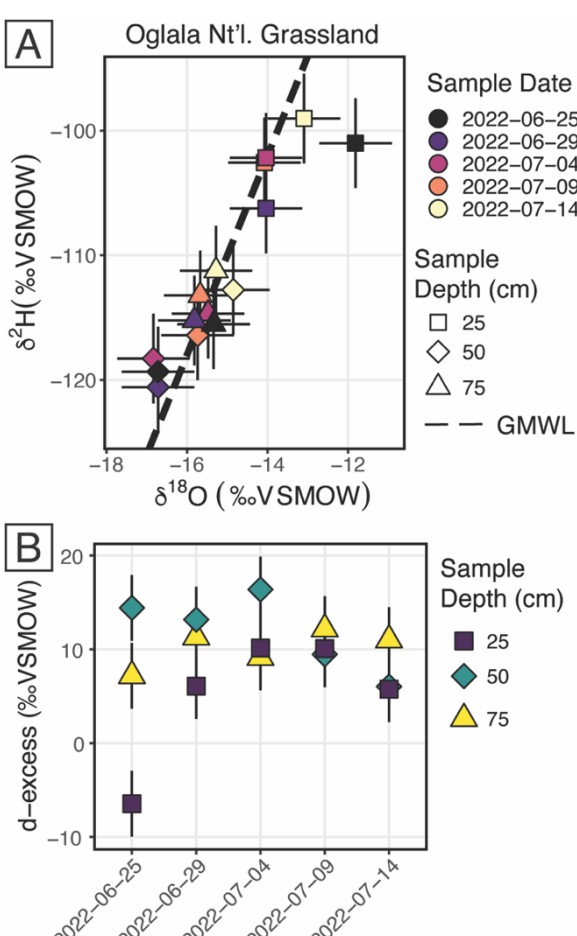

**Figure 8.** Results from the Oglala National Grassland, NE field site. A) $\delta^2H$ vs. $\delta^{18}O$, where the dashed
line is the global meteoric water line. The shapes for the different depths sampled matches figure 7, and
the color of the points is the date on which the soil water was sampled B) A plot of d-excess. Note, both
the color and shape match figure 7.

There are 10 samples from Briggsdale, CO (Fig. 7B, Table 3); five samples each from
vapor probes buried at 50 cm and 75 cm depth. Data from 25 cm at Briggsdale, CO were
excluded because the water vapor mole fractions from all of the flasks were extremely low
(<13,000 ppm). We excluded these data because these samples are associated with a very dry
soil (VWC < 0.05), and it is not clear how much sampling gas ($N_2$) is injected into the soil using
the vapor permeable tubing under very dry conditions (Quade et al., 2019), and therefore how
representative these isotope data are of soil water. Moreover, below 13,000 ppm there are large
linearity effects on a Picarro L2130-*i*, and it is challenging to correct those data if they were
measured using the dry-air carrier sample introduction method. While all samples overlap within
uncertainty for both $\delta^{18}O$ and $\delta^2H$ values, the absolute values of samples from 50 cm are
consistently offset to higher values for both $\delta^{18}O$ and $\delta^2H$ as compared to samples from 75 cm.
There are 12 samples from Seibert, CO (Fig. 7C, Table 3); four from each sampling depth
(25, 50 and 75 cm). At 25 cm depth, $\delta^{18}O$ values of three of the four samples overlap within
uncertainty, while the 25 cm sample from 2022-06-29 has a higher $\delta^{18}O$ value than the other
three samples. At 25 cm depth, $\delta^2H$ values overlap within uncertainty for all four samples. At 50
cm depth, there is a steady decrease in $\delta^{18}O$ value over the sampling period, while $\delta^2H$ values for
all four samples remain steady and overlap within uncertainty. At 75 cm depth, samples have a
very large range of $\delta^{18}O$ values between -8.5‰ and 7.4‰, and $\delta^2H$ values range between -
55.7‰ and 15.1‰. Almost all of the samples from 75 cm depth were associated with
condensation in the sample introduction lines during measurement.
**Table 3.** Results from the three field deployments of SWISS.

| Site | Date | Sample Depth (cm) | Flask | T (°C) | $\delta^{18}O$ (‰) | $\delta^{18}O$ (‰) Analytical Error | $\delta^2H$ (‰) | $\delta^2H$ (‰) Analytical Error |
|---|---|---|---|---|---|---|---|---|
| Briggsdale | 2022-07-17 | 50 | 3 | 25.1 | -10.8 | 0.2 | -65.6 | 0.6 |
| Briggsdale | 2022-07-17 | 75 | 4 | 23 | -12.1 | 0.2 | -69 | 0.7 |
| Briggsdale | 2022-07-22 | 50 | 6 | 25.9 | -10.7 | 0.3 | -67.1 | 0.7 |
| Briggsdale | 2022-07-22 | 75 | 7 | 23.6 | -11.9 | 0.2 | -69 | 0.6 |
| Briggsdale | 2022-07-27 | 50 | 9 | 24.3 | -10.4 | 0.3 | -65.6 | 0.6 |
| Briggsdale | 2022-07-27 | 75 | 10 | 23 | -11.5 | 0.2 | -67.6 | 0.7 |
| Briggsdale | 2022-08-01 | 50 | 12 | 23.4 | -10.7 | 0.2 | -67 | 0.7 |
| Briggsdale | 2022-08-01 | 75 | 13 | 22.4 | -12.0 | 0.2 | -69.1 | 0.7 |
| Briggsdale | 2022-08-06 | 50 | 15 | 24 | -10.5 | 0.2 | -65 | 0.6 |
| Briggsdale | 2022-08-06 | 75 | 16 | 22.9 | -12.1 | 0.2 | -68.8 | 0.7 |
| Seibert | 2022-06-19 | 25 | 2 | 24.2 | -8.3 | 0.2 | -59.8 | 0.6 |
| Seibert | 2022-06-19 | 50 | 3 | 22 | -7.8 | 0.2 | -57.8 | 0.6 |
| Seibert | 2022-06-19 | 75 | 4 | 19.4 | 7.4 | 0.2 | -7.6 | 0.7 |
| Seibert | 2022-06-24 | 25 | 5 | 24 | -8.7 | 0.2 | -58.7 | 0.7 |
| Seibert | 2022-06-24 | 50 | 6 | 22.2 | -7.9 | 0.2 | -56.7 | 0.7 |
| Seibert | 2022-06-24 | 75 | 7 | 20.5 | 4.9 | 0.2 | 15.1 | 0.6 |
| Seibert | 2022-06-29 | 25 | 8 | 23.2 | -7.4 | 0.2 | -56.9 | 0.6 |
| Seibert | 2022-06-29 | 50 | 9 | 21.8 | -9.1 | 0.2 | -56.7 | 0.7 |
| Seibert | 2022-06-29 | 75 | 10 | 21 | -5.6 | 0.2 | -42.1 | 0.6 |
| Seibert | 2022-07-04 | 25 | 11 | 25 | -8.7 | 0.2 | -60.6 | 0.7 |
| Seibert | 2022-07-04 | 50 | 12 | 23.3 | -9.9 | 0.2 | -58.8 | 0.6 |
| Seibert | 2022-07-04 | 75 | 13 | 21.5 | -8.5 | 0.2 | -55.7 | 0.7 |
| Oglala Ntl. Grassland | 2022-06-25 | 25 | 2 | 23.0 | -11.8 | 0.2 | -101 | 0.7 |
| Oglala Ntl. Grassland | 2022-06-25 | 50 | 3 | 22.8 | -16.7 | 0.2 | -119.3 | 0.7 |
| Oglala Ntl. Grassland | 2022-06-25 | 75 | 4 | 21.5 | -15.3 | 0.2 | -115.5 | 0.8 |
| Oglala Ntl. Grassland | 2022-06-29 | 25 | 5 | 25.0 | -14 | 0.2 | -106.2 | 0.7 |
| Oglala Ntl. Grassland | 2022-06-29 | 50 | 6 | 22.8 | -16.7 | 0.2 | -120.6 | 0.7 |
| Oglala Ntl. Grassland | 2022-06-29 | 75 | 7 | 21.3 | -15.8 | 0.2 | -115.2 | 0.7 |
| Oglala Ntl. Grassland | 2022-07-04 | 25 | 8 | 25.0 | -14 | 0.2 | -102.2 | 0.7 |
| Oglala Ntl. Grassland | 2022-07-04 | 50 | 9 | 23.0 | -16.8 | 0.2 | -118.3 | 0.6 |
| Oglala Ntl. Grassland | 2022-07-04 | 75 | 10 | 22.0 | -15.5 | 0.2 | -114.7 | 0.6 |
| Oglala Ntl. Grassland | 2022-07-09 | 25 | 11 | 23.0 | -14.1 | 0.2 | -102.6 | 0.6 |
| Oglala Ntl. Grassland | 2022-07-09 | 50 | 12 | 22.8 | -15.7 | 0.2 | -116.4 | 0.7 |
| Oglala Ntl. Grassland | 2022-07-09 | 75 | 13 | 22.0 | -15.7 | 0.2 | -113.2 | 0.6 |
| Oglala Ntl. Grassland | 2022-07-14 | 25 | 14 | 23.0 | -13.1 | 0.2 | -99 | 0.6 |
| Oglala Ntl. Grassland | 2022-07-14 | 50 | 15 | 22.8 | -14.9 | 0.3 | -112.8 | 0.7 |
| Oglala Ntl. Grassland | 2022-07-14 | 75 | 16 | 22.0 | -15.3 | 0.2 | -111.2 | 0.7 |


## 6. Discussion

### 6.1 QA/QC and field suitability tests

#### 6.1.1 Dry Air tests

In Colorado, where these tests were completed, the ambient atmosphere during the summertime typically sits at a water vapor mole fraction between 10,000 - 20,000 ppm, and in winter the water vapor mole fraction can drop as low as 4000 ppm. If the flasks had been slowly equilibrating with the atmosphere, the flasks would have drifted to much higher water vapor molar fractions. If the flasks did not drift towards higher water vapor mole fractions, we felt confident that the flasks are resistant to atmospheric intrusion after they have been flushed with dry air. We chose a timescale of seven days for the dry air tests because we found that in a low-humidity environment, seven days was enough time to meaningfully observe leaks, while being short enough to work through the QA/QC process efficiently. For example, results of two sequential dry air tests on the SWISS unit Toblerone (supplemental Fig. 2), show that it is possible to drastically reduce leaks that allow ambient water vapor in the air from intruding into the flasks  by tightening and/or replacing problematic fittings (both those attached to the glass flasks and those on the Valco valve) and in some cases the glass flask itself. During the final seven-day dry air tests, most flasks maintained a water vapor mole fraction less than 400 ppm, and all flasks maintained a water vapor mole fraction of less than 700 ppm (Fig. 3).

Across all of the SWISS units, there is a bias towards a higher water vapor mole fraction for the first flask that is measured (port one on every valve is the flask bypass loop, so the first flask is flask two), which suggests a methodological source of higher water vapor concentration rather than Swagelok fitting tightness problems. There are two potential sources for this issue. First, it is possible that not all of the atmospheric water vapor was flushed from the line that connects to the CRDS prior to the start of the measurements, but by the time the second flask is measured, the lines between the SWISS and CRDS have been sufficiently flushed, creating bias in the first flask measured.  This hypothesis could be tested by flushing all of the gas lines with dry air to progressively lower water vapor mixing ratios prior to measuring any flasks, to see what minimum ratio is required to eliminate this bias. Lab protocols can then be adjusted to flush all gas lines to this level. Similarly, it is possible that during the filling phase, not all of the atmospheric vapor has been flushed out of the Drierite system before starting the fill process. This hypothesis is supported by the systematic decrease in water vapor mole fraction across flasks in the Toblerone unit (Fig. 3, right panel). As a result of these biases, we now flush the Drierite for at minimum 30 minutes prior to the start of the experiment.

In addition to testing the leakiness, the dry air test also provided a useful baseline from which to test building materials. For example, in supplemental figure 5, we show the results of sequential seven day and 27-day dry air tests where we replaced stainless steel tubing and fittings with PTFE Swagelok fittings with 1/8 inch PTFE tubing. We thought that PTFE fittings would be advantageous because they are much easier to install and are significantly lighter, and would therefore be helpful when there are weight constraints. However, based on the very limited testing we did, PTFE fittings and tubing *may be* sufficient to store water for up to a single week, but on longer timescales (e.g. 27 days) we observed greater exchange and leaking than with the stainless steel fittings. We encourage any future user using this modification to rigorously test these fittings on a timescale appropriate for their application.

### 6.1.2 Water vapor tests

Our initial goal with the water vapor tests was to test whether the measured water vapor isotope values at the end of the two-week holding period were normally distributed about 0 within the uncertainty limits of the water vapor probes (Oerter et al., 2016). This was a reasonable goal given the similarities in probe set-up and the plumbing design between the SWISS and the IsoWagon system. But, the most salient result of the water vapor tests is that there is a consistent positive offset between the input isotope values and the isotope values measured at the end of the two-week experiments (Figs 4B, 5B). The positive offset in both $\delta^{18}O$ and $\delta^2H$ values is consistent across 11 different tests, using six different SWISS and three different input water isotope values. If there was alteration of original values due to leaky flasks, we might expect the $\delta^{18}O$ and $\delta^2H$ values to converge on the $\delta^{18}O$ and $\delta^2H$ value of the atmosphere. For example, we might expect water vapor from the light water test to have the most significant change in isotope value, towards that of the ambient atmosphere. Instead, the consistency across >135 flasks, different starting water vapor isotope values, sample introduction methods, and multiple analytical sessions suggests that this difference is a function of the storage and measurement process. In particular, the normality of the distribution suggests whatever the origin of the offset is, there is a systematic bias that we can reliably correct for.

### 6.1.2.1 Offset correction

To correct our data for this offset, we chose to use the median value as an offset correction rather than the mean of the normal distribution, because the median is not biased by major outlier isotope values that reflect abnormal values that go beyond analytical noise, such as a slow but major leak that changes the values far beyond the basic offset seen in the dataset. The calculated average offset is 1.0‰ and 2.6‰ for $\delta^{18}O$ and $\delta^2H$, respectively. After applying these values as an offset correction to the data, most flasks also fall within the uncertainty of the water vapor permeable probes ($\delta^{18}O = \pm0.5‰$ and $\delta^2H = \pm2.4‰$, Oerter et al., 2016), and the values are distributed about 0 (Figs. 4C, 5C). However, the uncertainty of the SWISS system is higher than that of the probes alone. Based on the results of the water vapor tests, we estimate the uncertainty of the SWISS at $\pm0.9‰$ and $\pm3.7‰$ for $\delta^{18}O$ and $\delta^2H$, respectively using the interquartile range (IQR) of the water vapor test results after removing outliers from the dataset. We prefer the IQR over the calculated standard deviation of the normal distribution, because IQR is not biased by outlier values. This level of uncertainty is large relative to other methods, but is sufficient for many critical zone applications, given the magnitude of seasonal variability in the top ~50 cm of a soil profile that can be observed in natural systems (e.g. Oerter et al., 2017; Quade et al., 2019). We also expect that uncertainties will decrease with future lab-based or near research facility testing and by comparing the SWISS against other soil water extraction methods.

The relationship between $\delta^2H$ values and $\delta^{18}O$ values in a dual-isotope plot provides insight into the mechanism driving the offset. Without an offset correction applied, the slope of the relationship between $\delta^2H$ and $\delta^{18}O$ is 3.14 ($R^2 = 0.62$) (Supplemental Fig. 4). This slope is only slightly higher than evaporation under pure diffusion (Gonfiantini et al., 2018). This suggests that the offset is likely driven by diffusion and will likely vary according to climate of the lab. For example, in a dry climate like Colorado, the water vapor concentration in the flask is significantly higher than the atmosphere, creating a larger diffusive gradient potential than for a lab in a more humid climate. We therefore strongly encourage future users to test their SWISS under climate conditions similar for their applications. Further, we encourage users who might use the

SWISS as part of a tracer study that uses labeled heavy water to test the SWISS with labeled
waters prior to their field experiments to verify reliability.
*6.1.2.2 Comparing sample introduction methods*
Supplemental figure 6 shows a kernel density estimate plot of the results from two water
vapor test sessions, with the offset correction applied. During the March 2022 session, flasks
were measured using the dead-end pull sample introduction method and during the August 2022
session, flasks were measured using the dry air carrier gas sample introduction method. There is
no significant difference in the measured difference between the two sample introduction
methods. That said, we prefer the dry air carrier gas method, because it is far simpler to control
the water vapor mixing ratio, and optimize the concentration to be around 25,000 ppm, which is
the concentration at which the Picarro L2130-*i* is most reliable. The dry air carrier gas method
also makes it easier to control for and monitor for condensation in the stainless-steel tubing and
vapor impermeable tubing, which can bias a measurement.
**6.1.3 Field suitability tests**
The long dry air tests in the field are a useful complement to the shorter in-lab tests
because they test the reliability of the system at field-deployment timescales. It is clear from the
34 and 43 day tests that the flasks are reasonably resistant to leaks on the timescale of a normal 4
– 6 week deployment  (Fig. 6A). These tests also give us confidence that flasks filled later in the
sampling sequence do not take on an atmospheric signal prior to sampling. There are a few
possibilities to explain the poorer performance of the Toblerone SWISS unit during the 52-day
test. (Fig. 6A). The first is that there is a real threshold past which the SWISS are no longer able
to retain samples. However, this explanation would suggest that there should be a gradual
decrease in performance across the three tests, which we do not observe. The alternative
explanation is that the poor performance is a result of inter-unit variability. The 52-day test was
the first long-term test and was performed in August 2021. In August 2021, we were continuing
to build new SWISS units and continuing to learn from each successive round of QA/QC, so it
seems plausible that there were unidentified problems with the SWISS unit Toblerone that were
solved before the water vapor tests in August 2022.
In figure 6B, the data show that the flasks preserved the $\delta^{18}O$ value of both flash-
evaporated and atmospheric water vapor over a seven-day period. One flask was removed from
the dataset (flask eight), because there was visible condensation in the clear impermeable tubing
during the measurement phase, with an increase of > 5‰ for $\delta^{18}O$ during the measurement
period. The condensation appeared as small (<1 mm) bubbles of water all along the impermeable
tubing, but the bubbles were concentrated near the connection between the SWISS and the
impermeable tubing. Notably, the two flasks whose $\delta^{18}O$ values do not overlap within
uncertainty are more negative than expected, rather than drifting towards atmospheric values or
values expected from diffusive fractionation. In contrast to the $\delta^{18}O$ values, only 3 flasks filled
with flash evaporated water vapor overlap within uncertainty of the known $\delta^2H$ values, while
four of the five flasks overlap within uncertainty of the estimated atmosphere isotope value. The
flasks tend to drift towards the value of the atmosphere, but retain the overall data pattern from
the oxygen isotope values.
The relatively high failure rate of this 'mock' field test was somewhat surprising given
the results of the water vapor tests done in the laboratory. Going into the test, we suspected that
flasks six and eight were slightly leaky based on previous water vapor tests; these were flasks
that previously performed poorly, but did not 'fail' during the water vapor test. Once we
collected the data, we compared the data for flasks six and eight to other flasks in the sequence.
During the measurement of flask eight, we observed condensation in the sample introduction
lines, and because the isotope values were so different relative to other flasks, we felt confident
in our exclusion of flask eight Flask six had $\delta^{18}O$ and $\delta^2H$ values similar to others from the
same sampling source, and seemed to fall within the pattern as expected. Therefore, we chose to
keep this data point in the dataset.
We hypothesize that one major problem with the mock field test dataset was the creation
of condensation in the sampling lines, as others have experienced in their setups (e.g. Quade et
al., 2019; Kühnhammer et al., 2019). Of particular interest are the flasks that had a lower than
expected $\delta^{18}O$ value (flasks four and nine). It is possible that those samples were also affected by
condensation, but in contrast to flask eight, which was excluded because of condensation during
measurement, we think that these samples may have been altered because of condensation at the
sampling stage. During condensation, we expect that $^{18}O$ will preferentially enter the liquid
phase, and that the water vapor that enters the flask will have a lower than expected $\delta^{18}O$ value.
The unique advantage of the SWISS is that it can operate independently, but with that comes the
trade-off that we cannot currently observe condensation in the lines during sample collection. To
prevent condensation from forming, other users have warmed the impermeable tubing between
the probes and the Picarro. The 'mock' field test data suggest that in many situations it may be
worthwhile to warm the transfer tubing, but this should be done in a way that does not alter the
thermal structure of the soil, and in remote settings, can operate safely independently.
***6.1.4 Lessons learned and recommendations from the QA/QC and field suitability tests:***
Our QA/QC process was a relatively efficient way to test the soundness of the SWISS
units. Through the QA/QC process we were able to identify problems with units, and
appropriately address them before deploying units to the field. We strongly recommend that any
user deploying SWISS to the field to undertake the same, or similar, QA/QC process.
The dry air test is a time-efficient and low-cost method for identifying flasks that are
leaky and will not preserve the sampled water vapor isotope values. It is useful during the
building stage to identify fittings that need to be tightened or flasks that need to be replaced, and
therefore we recommend these tests as a required pre-deployment step for future SWISS units.
We found that it was most time and energy efficient to move onto the next level of QA/QC once
13 out of 15 flasks of a SWISS unit had passed the dry-air test, because frequently the remaining
two flasks still had relatively low water vapor mole fractions (i.e. 500 – 700 ppm), and we could
sufficiently tighten the fittings prior to the start of the water vapor tests for them to be successful.
The dry air test is a low time and expense burden that can also be used to monitor SWISS units
for normal wear-and-tear (e.g. a flask that cracked during transport) during deployment periods.
Therefore, to ensure that SWISS units continue to operate as expected, we also recommend that
dry air tests be done between field deployments on every SWISS unit. Lastly, we note that the
dry air test could be modified based on available equipment (for example, if an instrument is
available to measure trace atmospheric gases, that could be used instead).
Based on the results of the long, field dry air test, we recommend that the water vapor
storage time doesn't exceed 40 days for reliable results, or that the user undertake multiple dry
air tests with either lower concentration benchmarks or longer duration if deployments may
exceed 40 days.
Overall, the quality control and quality assurance as well as the field suitability tests
demonstrate that the SWISS units can retain the isotope values of water vapor collected using
water vapor permeable probes. Like many other systems that measure dual isotopes, each system
(i.e. $\delta^{18}$O and $\delta^2$H) must be evaluated separately. In general, we interpret oxygen isotope data
with a higher degree of confidence than the hydrogen isotope data. As the automation test
revealed however, even when the absolute $\delta^2$H value is not correct, the general pattern can reveal
information about soil water dynamics.
Finally, we opted to use a large flask volume because it allows us to measure a sample for
long enough on a CRDS that we get reliable data, without interacting with vapor bound to the
flask walls. The drawback of this, however, is that we must sample soil water vapor for a
relatively long period of time (45 minutes). In supplemental figure 7, we show that the sampling
regime, and particularly the length of time we pump dry air through the tubing, does not
significantly alter the soil moisture content of the soil. Additionally, we demonstrate that the
sampling regime we use does not introduce significant memory effects.
**6.2 Field Deployments**
In Figure 7 we show the results of three field deployments completed during summer
2022 (Table 3). At the Oglala National Grassland site, we used the SWISS unit named Lindt to
collect samples. During the August 2022 water vapor test on Lindt, all $\delta^{18}$O values fall within
uncertainty of the system, and nine of the fifteen $\delta^2$H values fall within uncertainty of the
system. Therefore, we interpret the $\delta^{18}$O values with greater confidence and the $\delta^2$H values with
lower confidence (Figs. 4C and 5C). We note that the $\delta^{18}$O and $\delta^2$H values broadly follow the
same trends, and fall on the global meteoric water line (Figs. 7 and 8A). In general, soil water
from 25 cm had higher $\delta^{18}$O and $\delta^2$H values than soil water from both 50 and 75 cm (Fig. 8A).
Given that 4 of the 5 samples from 25 cm overlap with the GMWL and have a d-excess that
overlaps with $10 \pm 2.6$‰, the soil water from that depth may reflect summer precipitation with
higher $\delta^{18}$O and $\delta^2$H values. Soil water from 75 cm had intermediate $\delta^{18}$O and $\delta^2$H values for
most of the study period, and soil water from 50 cm depth had the lowest $\delta^{18}$O and $\delta^2$H values
for most of the study period, which may reflect a more mean-annual or winter precipitation
biased value. Based on data available from the National Weather Service (Chadron, NE), there
were likely significant precipitation events on 2022-06-25 and 2022-07-08 at the field site. There
is a significant shift to lower $\delta^{18}$O values at a sampling depth of 25 cm between 2022-06-25 and
2022-06-29, as well as a marked increase in the d-excess value (Fig. 8A). We interpret this shift
as infiltration of precipitation with lower $\delta^{18}$O values, which is supported by a return of d-excess
values to ~10‰ (Fig. 8A). The National Weather Service reported 21.33 mm (0.84 inches) of
rain at Chadron Municipal Airport, approximately 50 km from the study site on 2022-07-08,
which likely was associated with at least some precipitation at our field site. Following the
significant rain event on 2022-07-08, we observe a marked increase in the stable isotope value of
water vapor from a sampling depth of 50 cm, towards values that are much closer to those at 25
cm depth. These data suggests that soil water isotopes at 50 cm in this silt-loam Aridisol may be
fairly sensitive to large individual precipitation events, while at 75 cm soil water isotopes remain
comparatively uniform. Future work should address how drought conditions, storm size, pore
size distribution, and soil clay mineralogy influence the variability of soil water isotopes with
depth.
At Briggsdale, CO we used the SWISS named Raclette to collect soil water vapor
samples. Data from 25 cm depth at Briggsdale, CO were discarded because the water vapor mole
fraction was much lower than would be expected given the soil temperature (i.e. < 15,000 ppm).
The gravimetric water concentration (GWC) at that soil depth at the time of sampling was
approximately 4% through the sampling period. Future work should include a multiple-method
(e.g. cryogenic extraction, centrifugation, etc.) comparison of soil water isotopes at low water
contents to better understand what these samples might represent, and if they are actually
representative of soil conditions.
Based on the results of the August 2022 water vapor test done on Raclette where all
flasks fell within uncertainty of the SWISS system for both $\delta^{18}O$ and $\delta^2H$, except for flask 11
(Figs. 4C and 5C),  we interpret all data with greater confidence. Flask 11 corresponds to the 25
cm depth sample from 2022-07-27, and was already culled from the dataset because of low water
vapor mole fraction associated with the very dry soil. The soil water $\delta^{18}O$ and $\delta^2H$ values from a
sampling depth of 50 cm and 75 cm overlap within uncertainty, but the soil water $\delta^{18}O$ and $\delta^2H$
values from 50 cm are higher than the isotope values from 75 cm. All of the data from each
sampling depth group (i.e. 50 cm and 75 cm) overlap within uncertainty, conforming to the
expectation that soil water from these sampling depths should be fairly invariant (e.g. Oerter et
al., 2019). There were precipitation events at the study site on 2022-07-24, 2022-07-28 and
2022-07-31. It is possible that the slight negative shift in both $\delta^{18}O$ and $\delta^2H$ on 2022-08-01
reflects infiltration of precipitation to those depths, but this is not certain given that all of the
measurements from within a sampling depth overlap within uncertainty.
At Seibert, CO we used the SWISS named Toblerone to collect soil water vapor samples.
The soil water isotope data from 75 cm depth at this site offer a few useful lessons for future
users. The two key observations of the data from 75 cm depth are that the $\delta^{18}O$ and $\delta^2H$ values
are much higher than the other two sampling depths d , and that the  $\delta^2H$ and  $\delta^{18}O$ values do not
move in parallel with each other. While measuring these samples we observed condensation in
the impermeable tubing at the point where the SWISS connects to the impermeable tubing.
Additionally, when we heated the stainless steel tubing that connects the tubing flask and Valco
valve we observed a rapid increase in water vapor mole fraction (1000's of ppm over <30
seconds) that was accompanied by a rise in stable isotope value. During these measurements, we
were rarely able to get a stable isotope value measurement window, and instead the stable
isotope value of the vapor increased continually through the measurement. It is for these reasons
that we feel confident in discarding the stable isotope data from 2022-06-19 – 2022-06-29. The
final measurement from 75 cm depth on 2022-07-04 approaches a reasonable isotope value when
compared to isotope values from the other two depths, and that sample had fewer condensation
problems during measurement. However, because we have no sequential context for what a
reasonable value for this depth is, we discarded that value as well. For that final 75 cm sample,
we were more successful because we warmed the entire length the vapor impermeable tubing, as
well as the stainless-steel tubing, flask, and Valco valve evenly so that there were no temperature
gradients across the vapor path. If the condensation had only been in the impermeable tubing it
would have been much easier to successfully analyze these samples by just closing off the flask
and running dry air through the tubing to remove condensation, but because condensation was
also occurring in the stainless steel tubing between the flask and Valco valve, this was not
possible. It remains unclear why condensation was such a significant problem for samples from
that depth as opposed to samples from different depths in the same SWISS. Future work should
include further testing of the SWISS across different water contents and temperatures to better
understand why the phenomenon may have occurred.

Based on the results of the August 2022 water vapor test done on Toblerone, we interpret
all data from 50 cm and 25 cm depth with high confidence, except for Flask 3, which is the 50
cm sample from 2022-06-19 (Figs. 4C and 5C). Unlike data from the other two field sites, soil
water from 25 cm and 50 cm overlap within uncertainty. There were two precipitation events at
the field site during the sampling period on 2022-06-25 and 2022-07-01, but both events were
quite small (<0.5 mm, CoAgMet). There is no significant influence of the precipitation events on
the $\delta^{18}O$ and $\delta^2H$ values. The >1.0‰ increase in $\delta^{18}O$ values on 2022-06-29 is surprising given
that there is not a comparable magnitude increase in $\delta^2H$ value, and that the values measured
from 2022-07-04 more closely match the $\delta^{18}O$ and $\delta^2H$ values from the two earlier sampling
days. There are two potential explanations for this data. First, that this shift is a real signal from
an evaporation driven increase in the $\delta^{18}O$ value, and the shift back to a lower $\delta^{18}O$ value on
2022-07-04 is due to the infiltration of precipitation, which could also explain the low d-excess
value associated with this measurement (Supplemental Fig. 8 ). The second possible explanation
is that the 25 cm sample from 2022-06-29 is influenced by condensation at the time of sampling.
Dew point at the field site on 2022-06-29 significantly decreased as compared to the other
sampling days to a monthly minimum of 20.6°C (CoAgMet). It is possible that environmental
conditions encouraged the formation of condensation in the impermeable tubing at the time of
sampling. There were no obvious signs of condensation during the time of measurement in the
lab. These results highlight the utility of having broad contextual environmental data to aid in the
interpretation of soil water isotope data.
All together, these three soil water isotope datasets demonstrate two main findings. First,
data from these samples show that the differences between field sites are easily resolvable using
the SWISS. For example, at 50 cm depth the oxygen isotopes range between -14.4 to -16.3‰, -
9.9 to -10.3‰, and -7.4 to -9.3‰ for the Oglala, Briggsdale and Seibert sites, respectively. These
differences likely reflect differences in the stable isotope composition of precipitation and
infiltration and evaporation dynamics. Second, the sample data retrieved from a SWISS are
sufficiently precise to be able to meaningfully resolve vertical profile soil water isotope data. For
example, at the Oglala National Grassland field site, soil water from 25 cm clearly has higher
$\delta^{18}O$ and $\delta^2H$ values as compared to soil water from a depth of 50 and 75 cm.
**6. 3 Future improvements and future work**
One significant SWISS unit hardware improvement that could be made would be to
install a heating implement to the flasks. One source of uncertainty on the current system is the
potential effect of uneven heating of the flasks prior to measurement which may create
temperature gradients that are large enough to allow for condensation when warm vapor meets a
slightly colder spot. This could be improved in subsequent iterations of the SWISS with the
addition of heat tape or blankets that can deliver controlled heat and create consistent
temperatures. This improvement would also help limit the amount of manual intervention needed
during measurement, and could improve automation of flask measurement. Additionally, finding
a way to safely and automatically heat the impermeable tubing that connects the water vapor
probes and the SWISS in a way that doesn't change the inherent thermal structure of the soil, and
is safe for unmonitored use, would help to prevent the formation of condensation in the field and
reduce the uncertainties related to sampling.
We have made a few improvements to the automation system that were not implemented
for the data presented in this contribution, but will be part of future deployments. First, we will
track conditions inside the SWISS with a temperature and relative humidity sensor inside the
case. Second, we plan to eliminate the power inverter by powering both the Valco valve and
mass flow controller with VDC using a power step up controller. Lastly, we will add an IoT
cellular router to be able to remotely monitor and control the SWISS units. This would be
particularly helpful if there is a sampling day that is unexpectedly cold or when the dew point at
the field site is unexpectedly low and we expect condensation to form more readily form in the
field, or if there is a precipitation event that we are interested in capturing, because with the IoT
cellular router we could remotely alter the sampling plan.
While the improvements and additional testing we have done to the SWISS in this
contribution represent a significant step forward, additional work should be done to make the
system more useable by the ecohydrology community. We have rigorously tested the SWISS in
the lab, and demonstrated a few ways in which the SWISS can fail in field settings. A full
comparison of how soil water isotope data collected using a SWISS as compared to other in situ
(both vapor probes and lysimeter) and destructive sampling methods would shed light on the
accuracy and precision of our system, and the applicability of our lab-based experiments to the
field. These experiments should be carefully designed with considerations of soil grain size, soil
water content, expected isotope values, and climate. Additionally, we plan to test SWISS unit
resilience during air travel so that these units can be used at field sites that are not within driving
distance of a research facility.

## Conclusions

We presented the evolution of the soil water isotope storage system (SWISS) from a
prototype to a fully built out and tested system. We also presented a quality control and quality
assurance procedure that we strongly recommend future users undertake to ensure the reliable
storage of soil water vapor over long time periods (up to 40 days). In addition, these quality
control and quality assurance tests shed light on the accuracy and precision of the SWISS. After
applying an offset correction, we determine the precision of the SWISS to be ±0.9‰ and ±3.7‰
for $\delta^{18}O$ and $\delta^2H$, respectively. In a field setting, flasks reliably resist atmospheric intrusion.
Additionally, the proposed sampling schema does not introduce significant memory effects.
Lastly, we demonstrate that the current precision of the SWISS still allows us to distinguish
between field sites and between soil water dynamics within a single soil column. Taken as a
whole, these data show that the SWISS can be used as a tool to answer many emerging
ecohydrological questions, and will enhance researchers' ability to collect soil water isotope
datasets from more remote and traditionally understudied field sites.

## Acknowledgements

We thank the numerous field assistants who helped to make the field work presented in
this paper possible, including Spencer Burns, Anne Fetrow, Sarah Brookins, Juliana Olsen-
Valdez, and Haley Brumberger. We acknowledge that both field work and laboratory work for
this study were done on the traditional territories and ancestral homelands of the Arapahoe, Ute
and Cheyenne peoples. This work was supported by startup funding from CU Boulder and NSF
funding from grant EAR-2023385 awarded to K. Snell. Additionally, this work was supported by
the University of Colorado Boulder Beverly Sears Research Grant and the Clay Minerals Society
Graduate Student Research Grant both awarded to R. Havranek. CUBES–SIL is a CU Boulder
Core Facility associated with RRID: SCR_019300.

**Author contribution**

Rachel E. Havranek: conceptualization, methodology, investigation, formal analysis, funding acquisition, writing – wrote original draft, review and editing. Kathryn E. Snell: conceptualization, methodology, writing – review & editing, funding acquisition. Sebastian H. Kopf: conceptualization, methodology, writing – review & editing. Brett Davidheiser-Kroll: conceptualization, cethodology, writing – review & editing. Valerie Morris: methodology, writing – review & editing. Bruce Vaugh: methodology, writing – review & editing.

**Competing interests**

The authors declare no competing interests.

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
