# Peer review of "Technical Note: Lessons from and best practices for the"

_EGUsphere, 2022_

## Author Response (AR1)

**Reviewer #1 (Stefan Seeger):**
The manuscript "Lessons from and best practices for the deployment of the Soil Water Isotope Storage System" presents an important step towards the automated acquisition of in-situ soil water vapor samples from remote sites that are not suited for the field deployment of stable water isotope analyzers. The authors have submitted a very well structured paper that I enjoyed reading.

1. My biggest concern is of formal nature, because in my opinion this paper is more a technical note than a research paper - I do not see any research question asked or answered, nor do I see any scientifically meaningful findings. Nevertheless I really do appreciate the effort that has been made to thoroughly develop and test such a setup and I think the manuscript is worthy for publication.
We agree that this paper is more of a technical note rather than a research paper and will change the submitted format appropriately.

2. Concerning the actual content of the manuscript, I would appreciate a more detailed discussion of the outliers and observed system failures.
2.1. In Line 240, you describe that you repeated your dry air tests "until the majority (at least 13/15) flasks" had a sufficiently low moisture content. Why do not all flasks have to pass your threshold criterion. How do you treat measurements of the somewhat leaky flasks?
2.2 During your testing with known isotopic compositions it is easy to spot faulty flasks, but on what basis would you treat actual field measurements? Is there a reliable way to spot outliers?
2.3. During the field application you had to discard 10 of 45 measurements. That is a failure rate of over 22%. Do you have any suggestions on how to reduce this failure rate?

Because the goal of this paper is to make the system as usable by the greater community as possible, and to be as transparent as possible about the system's strengths and weaknesses, we are happy to add further descriptions of the outliers and system failures.
We appreciate comment 2.1 and more clearly articulated our philosophy about how to move through QA/QC efficiently; our threshold of at least 13/15 flasks holding dry-air was to be able to keep moving through the process so that SWISS units could be ready for field deployment. This discussion is now also included in section "6.1.4 Lessons learned and recommendations from the QA/QC and field suitability tests." There, we noted ways in which we would likely do QA/QC differently in the future, so that other users can adopt a QA/QC process that works for their uses.
We used the 'mock' field deployment section to discuss in greater detail ways to identify if condensation in the sample introduction lines is affecting isotope values during measurement based on peak shape, and ways we identified suspected condensation during vapor collection. We also used the mock field test section as a place to discuss more thoroughly how we treated flasks that we suspect are slightly leaky. Lastly, we also discussed the importance of sequential data collection to identify data that are spurious, and our approach to designing sequence timing to be able to both identify bad samples and answer the question at hand.
A 22% failure rate is quite high, but not unexpected for this testing phase of development; part of the point of this paper is to help others who use this in the future deploy

the SWISS with a better success rate. Our field deployment points to ways that either the automation system can be improved and/or when the SWISS is likely to fail. First, it's not entirely clear what soil water vapor samples collected from very dry soils using the vapor permeable tubing represent, and a targeted study looking at dry soil conditions would be a really helpful next step. Second, some updates to the automation schema that make it more remotely controllable would help researchers avoid sampling on days when condensation in the field might be likely.

Apart from that, I only have minor points to comment on.
Line 73: Your list of field deployments of laser-based instruments for in situ measurements is missing some notable cases that either are older or produced much more measurements than your given examples:

Volkmann2016 https://doi.org/10.1111/nph.13868

Gaj2016      https://doi.org/10.5194/hess-20-715-2016

Seeger2021   https://doi.org/10.5194/bg-18-4603-2021

Gessler2022  https://doi.org/10.1111/nph.17767
We've added these references, thank you very much for the suggestions

Line 267-268: Could you elaborate on why and how the heating does "create a longer and more stable window of measurement time". I kind of get a vague idea about this within the following paragraph, but I think a more explicit explanation of the effect of this heating procedure right at this point would be suiting.
We added the following text to explain the purpose of the heating: "The purpose of the heating was to create a longer, more stable window of measurement time. Heating the flasks creates a more stable measurement by limiting water vapor bound to the glass walls of the flask and by helping to homogenize the water vapor within the flask."
In general, this practice was something we found somewhat by accident, and seems to help, but it may not be required.

Lines 285-286: "For some flasks, using either a later portion of the measurement period, or slightly earlier offered a more stable signal." Do you treat those bottles differently during normal operation of the system, or do you treat all bottles equally, even though they might perform better if treated more individually?
During normal operation we took care to monitor each flask as it was measured and flagged any flask that needed individual attention during the data reduction process. There may be some pattern to which flasks needed individual attention, but in general it just depended on conditions at the time of measurement.

Lines 318-319: You refer to a "SWISS unit offset correction (detailed below)". Then there is one last method subsection and the results begin. You kind of describe an offset correction in the results subsection "5.1.2. Water vapor test", but maybe the description should happen within the methods part of the paper? Or you could place a reference to where the offset correction is actually described.

To clarify what we meant by the offset correction we created a specific subsection of 6.1.2 so that it is easier to find our discussion of that correction, and added reference to that subsection in the methods section.

Lines 349-370: In that section you describe 18O and 2H completely separately. Do the outliers of 18O coincide with the outliers of 2H, or are they completely independent? How do the outliers look like in the dual isotope plot? Could that help to diagnose why these outliers are outliers? This is a really important point, so we added a short paragraph in the results section comparing figures 4C and 5C, and have added reference to the SI dual isotope figure. We also added text to the discussion section 6.1.2.1 comparing the slope of the dual isotope plot with slopes expected for either entirely diffusive (kinetic) fractionation or entirely equilibrium fractionation.

Lines 374-380: I think the long term field storage tests look a lot like 43 days of storage are not any worse than 34 days. Do you expect that there is actually a critical time limit somewhere between 43 and 52 days which can explain the difference between Fondue and Toblerone? Or has Toblerone issues that also would affect its results after a shorter storage period?

Repeated tests with one and the same SWISS unit over different time spans might be more informative than the comparison of three separate units after different storage times. (But of course that would take a lot of time...)
    In this instance, we suspect that the difference between 43 and 52 days is likely more controlled by unit to unit variability or the conditions under which the SWISS units were deployed. There may be some specific turn-over point that would be universally applicable to different SWISS units under different deployment conditions (it is entirely possible that SWISS could be used to store water vapor on longer timescales), but we chose to share these data to demonstrate that any user who hopes to use longer deployments should carefully test their SWISS in their climate.
    We streamlined and clarified text in discussion sections 6.1.3 and 6.1.4 to help make some of these points clearer for the reader.

524: "Samples were taken approximately every five days...". Why is it "approximately" five days? I suppose your automation procedure does not involve a random number generator? Shouldn't you be able to state the precise sampling interval? Maybe just drop the "approximately" (even if - for whatever reasons - it weren't perfect 5-day intervals). Thank you for catching that, we removed the word approximately.

Line 218: Could you specify the type and manufacturer of the "helium leak detector". We used a Restek Leak Detector, and added that information to the manuscript.

Line 247: "...did not lead by the time..." maybe you meant "leak" instead of "lead"? Thank you for catching this! Fixed.

**Reviewer #2:**
**General comments:**
In their manuscript entitled "Lessons from and best practices for the deployment of the Soil Water isotope Storage System", Havranek et al. describe their experience with automating and operating a self-developed soil water vapor sampling system for subsequent, lab-based stable isotope analysis. They present an extensive testing procedure necessary prior to unattended field-deployment. Overall, the manuscript contains a lot of detailed instructions and hints that are definitely helpful for replicating the setup and producing soil water isotope datasets, which are valuable for the ecohydrology community and thus the readers of HESS. Therefore, I appreciate the effort of the authors and I recommend publication after major revision following the comments I specified below.

The authors thank this reviewer for their thorough and constructive review. Their comments significantly improved this manuscript. Below we detail our response to their comments.

In my opinion, the manuscript would benefit from a better structuring as currently some important method details are described in the result section, part of the data is presented not before the discussion section etc. (see details in the specific comments). Also, the discussion section currently comprises a lot of repetitions of method and result details. Instead, it should be focused more on the critical evaluation of the presented data and other findings. Given the problems encountered in the field and the corresponding failure rate, I find it unfortunate to call it a "full" testing as obviously not the full spectrum of potentially relevant environmental impacts could be accounted for during those tests.

We appreciate the concerns about the structure of the paper – we found it challenging to structure the paper in such a way that clearly articulates the sequential development steps of the SWISS, while maintaining a traditional scientific paper format. In revisions, we more clearly separated results from the discussion, and appreciate the reviewer's help identifying specific places to streamline the discussion.

Further, I would have liked to see an independent validation of the presented soil vapor isotope data by established means in order to have a profound accuracy assessment. In my understanding, precision is the standard deviation of replicate measurements, which have not been performed on soil samples. Therefore, I am not sure whether the precision and accuracy assessments from the lab tests can be assumed for the natural soil samples as well.

We appreciate that the data we have presented is not truly a 'full' testing of the system, and we did not design these initial experiments to test the SWISS against other methods of extraction soil water isotope data, nor is this paper intended to address that stage of development and testing. Our goal with this paper is to get the SWISS out quickly to other potential users so that they may be able to build and test the SWISS for their scientific questions (which might include some of these environmental tests). We agree that testing the SWISS against other methods of soil water extraction is an important next step for testing the SWISS, but requires a more focused experiment that fits the needs of multiple stakeholders in the ecohydrology community. Further, the first-order tests that this paper presents were a necessary first step to be confident that the system works as expected, and to learn how to identify problems, so that later environmental experiments will be reliable. That level of testing should be done to complement the testing that has already been done on the vapor permeable

tubing and to address concerns about the reliability of the tubing across different soil textures and environments. It is therefore outside the scope of this paper to complete that level of testing, though we agree that is an important and obvious next step. In revisions, we adjusted how we describe the level of testing that has been done on the SWISS and advocate more clearly for rigorous testing of the system against other traditional methods of soil water extraction methods in our future work section.

Formally, i would recommend using italics (once introduced) for the different methods throughout manuscript, clearly distincting between method, results and discussion statements, and using past tense in method and results.

We appreciate this comment and edited the methods sections so that the different methods are more clearly defined. We also carefully edited the manuscript to correct tense and grammar.

**Specific comments:**

Line 12: "full" may be too strong given the unaccounted-for circumstances encountered during the unattended field-deployment

That's fair, especially given the future work we now more clearly advocate for in the discussion. We have removed "full".

L19: You are stating the precision. What about the accuracy? Do you assume perfect accuracy even for non-lab conditions after performing the offset correction?

Here we don't intend this accuracy as perfect accuracy, but accuracy relative to known standards. We can't spell this out in detail in the abstract, but in the main part of the text that deals with this issue, we take care to describe the nature of the offset correction that shifts our raw data into the VSMOW reference frame that other labs also correct to. In that sense, this correction is analogous to common standard correction procedures of all stable isotope methods that assume each analytical set up has its own unique set of isotope effects on their data that correction with known standards accounts for. So, we have left accuracy and precision in this place in the abstract given that the main text spells out this process.

L22: "faithfully" may be too strong given that soil water isotope data were not validated by established means.

We have removed the word faithfully.

L31: root WATER uptake

Thank you. Fixed

L40: You might consider mentioning here also the work of Wassenaar and colleagues (doi: 10.1021/es802065s) who invented the principle of using laser-based instrument to measure vapor for liquid water isotope assessment. All in situ isotope sampling approaches are now based on this principle.

This is a great suggestion; we've added this reference. Thank you.

L46: The expansion OF in situ…

Thank you, added

L50-54: I am sure this is a good example to prove your point why a setup like yours is helpful/needed. Unfortunately, I do not understand what it means. What are br-GDGTs?

Branched glycerol dialkyl glycerol tetraethers are lipid membranes synthesized by archaea and bacteria in soils. They are a very useful paleoclimate proxy because they don't degrade through time, but questions remain in the geologic community about how the geochemistry we measure today relates to the climate conditions (e.g., temperature and precipitation) that paleoproxies like bg-GDGT are used to reconstruct; studies of modern environments that tools like the SWISS help enable will improve these proxies. To help make this more understandable, we rephrased the sentences and expanded the original text.

L62-63: I would expect such recommendations to appear in the discussion/conclusion but not in the introduction.

We've removed this point from the introduction. And made it clear in the discussion that the QA/QC process should be used by all future users.

L68-96: This reads like a second introduction where you identify a problem you are intending to solve. Why not merge this information with section 1?

We appreciate this comment. We significantly condensed this text and merged with the final paragraph in the introduction.

L79: vapor CONCENTRATION gradient

Thank you. Fixed

L89: you already defined the abbreviation "SWISS" in line 61/62

Thank you, we removed the longer text

L103: Do you have indications/references that this time frame is sufficient for the soil to return to natural conditions? Can this be tested? How?

This is a good question. We are not aware of studies that have focused on this issue, although it may exist especially in the lysimeter literature. Most other studies that use the vapor permeable tubing typically install the probes 2 – 3 months ahead of time. The long timeframe of our deployment to measurement was based more on logistical constraints rather than concerns about soil settling. We've added some context about the settling time of our study compared to others in the text, and also briefly alluded to some of the logistics of why we installed probes so far in advance.

L121: Wouldn't you want to prevent condensation instead of only limiting it?

Yes, that's true, we've replaced the word limit with prevent

L140: This reference is missing in the Works Cited section

Thank you for catching this, we've added this citation

L144: Wouldn't vapor bound to the flask walls cause a memory effect that needs to be re-assessed and corrected for prior to every new deployment of each single flask? If so, how would this impact the universal offset you applied if not all flask had the same filling history in terms of isotopic composition prior to common deployment?

This is an important point. We added a point later in the methods to emphasize that before every single deployment, the flasks are cleaned by flushing thoroughly with dry air, effectively removing any prior isotope effect.

L147 and throughout MS: Please do not use different formats for stating the tubing diameter (1/8 inch vs. 1/8" vs. 1/8th inch etc.)
We've adopted the 1/8inch format throughout the manuscript, thank you for catching this.

L151: please define the material of the foam and the nature of the insulation (thermal?)
We've added more details about the Pelican case and the foam that comes standard with it.

L160: Do "offsets" mean that there is no equilibrium at shorter (or greater?) lengths?
It seems like the total surface area of the vapor permeable tubing affects the measured isotope values, and so at shorter lengths the effect is greater. But, the effect is consistent, and so what really matters is that the probes used for samples and standards are identical. Another research group uses 20 cm long probes (e.g., Quade et al., 2019).

L162: Was it mandatory as well to cut the Bev-A-Line connections to identical lengths?
We chose to construct the probes as identically as possible to limit the amount of potential variability that could stem from variable memory effect in the probes.

L186: Please define VDC
We've defined this term as well as others below.

L200: In my opinion, such statements discussing the suitability of and potential alternatives for certain components are better placed in the discussion
We have removed this text in this location and moved it to the discussion.

L218: Please define the helium detector model and manufacturer.
We used a Restek Leak Detector and have added that information to the manuscript.

L219: To me, this is a result statement
We rephrased this statement as more of a goal, rather than a statement of results.

L230: what quantity was measured on each flask? I understand that the dead-end pull method applies a vacuum to the flasks. Were the custom-made flasks tested for vacuum suitability? Are extra safety measures necessary for this step? Would a vacuum test be able to identify large, medium, or small leaks all at once?
    We measured the flasks for five minutes, resulting in ~150 ml of air being removed from the flasks. No extra safety measures were necessary for this step.
    A vacuum test is an interesting alternative that we didn't pursue. A vacuum test could certainly detect a large and medium leak at the same time, but we're not sure about the small leak. A small leak test should be done sequentially after the other two tests. Depending on the timescale, a vacuum line could likely be used for the small leak, because we have been able to see small leaks when we use the dead-end pull method and look at the outlet valve values as a proxy for pressure in the flask. However, any user employing this tactic should make sure that they are working at sufficiently high vacuum and holding it for long enough to really see leaks on a timescale appropriate for their application.

L231: please move the parentheses to line 227 and use italics consistently throughout the manuscript when naming the different methods
We have edited our methods to clarify the different methods and steps of each QA/QC step.

L235: "we found…" sounds like a result statement
We have moved this statement to the results section

L236:"…would likely…" sounds like a discussion statement
We have moved this statement to the discussion

L240: Why did the dry air tests have to be repeated? Did you encounter gradual decrease of vapor adsorption at the flasks' walls or did you change anything in the setup (e.g. tightening of connections) between those tests?
We responded to this comment in part in the methods section, but also added extra detail to the results and discussion sections. In the results section we make a more direct reference to SI Figure 4, where we tightened fittings between two successive dry air tests and see an improvement in water vapor mole fraction for most of the flasks of the Toblerone SWISS unit. As part of that discussion, we clarified that we tightened connections between each successive test.

L247: I think it must be "leak", not "lead"
Fixed, thank you

L253: How far and for how long was the tubing immersed in water?
We added the following text to add some clarification: "We immersed the probes up to the connection between the vapor permeable and impermeable tubing in water, taking care to not submerge the connection point and inadvertently allowing liquid water to enter the inside of the vapor permeable tubing." We also moved up the sentence: "We flushed the flasks at a rate of 150 ml/min for 30 minutes, and measured the $\delta^{18}O$ and $\delta^2H$ values and mole fraction of water vapor as each flask was filled."

L270: How was dryness verified?
We added the following text: "To verify that the impermeable tubing between the SWISS and the Picarro was sufficiently dried, we waited until the water vapor mixing ratio was below 500 ppm between samples."

L279: Picarro claims that the measurement range for their L2130-i model is 1000-50000 ppmv. How come that you have condensation issues at vapor concentrations below 50000 ppmv?
It is entirely possible to measure an isotope value up to 50000 ppmv, however, at both high and low concentrations (i.e., below 10,000 ppmv and above 40,000 ppmv) linearity corrections become much larger and have a non-linear trend. To correct values into an accepted reference frame at those concentrations requires that standards are run at multiple concentration steps (i.e., 36000, 37000, 38000, etc.).
We have added text into this paragraph that clarifies that large linearity corrections are needed outside of the optimal humidity range.

L281: Why does changing pressure cause isotope fractionation?
At the core of this issue are vapor pressure isotope effects, where the partial pressures of water vapor with 18O versus 16O evolve during condensation and evaporation. To make it a bit more clear how this relates to the different sample introduction methods, we added this sentence "It is possible that during a dead-end pull on the flask, that heavier isotopes may remain attached to the walls of the flask, coming off later as the pressure drops."

L283: what signal was supposed to be stable? Isotope? Vapor concentration?
We have added the following text: For each flask we looked at the stability of the isotope values as well as either a stable water vapor mole fraction if the dead end pull method was being used or a steady, linear decrease in water vapor concentration if the dry air carrier gas method was being used

L287: why was it necessary to use two different methods to assess instrument stability? And how stable was the instrument compared to the SWISS analytical uncertainty?
We have added text to the results section that addresses the Picarro uncertainty, which was typically d18O = 0.2-0.3‰, d2H = 0.6-0.7‰ for the three-minute data collection time. Over an analytical day, instrument drift was usually <0.05‰ for d18O and <0.3‰ for d2H. Both are well within the uncertainty of both the vapor permeable probes and SWISS units.
Use of the word stability in this instance refers to long term stability (i.e., hours) rather than stability of any single measurement. This was confusing, so we changed that sentence to clarify that we were assessing instrument performance and drift over the analytical day.

L295: In what aspects were field conditions different from lab conditions and why wasn't it sufficient to perform these tests just outside the lab building? Are the results of these tests not transferable to other field sites?
The goal was to put SWISS units in a representative field deployment situation, where they experience the daily temperature and relative humidity fluctuations that they would during a typical deployment. We added text to make that goal clearer. We also expanded on the applicability of our field tests to other locations in the discussion as well.

L314: Using two different methods within one batch of samples and standards contradicts the principle of identical treatment (Werner & Brand, 2001, doi: 10.1002/rcm.258). Why is it justified to do so in this case?
We added text to clarify that all 15 samples were collected using the vapor permeable probes, so all 15 samples were taken in an identical fashion. We chose to collect atmosphere alongside two waters in this instance to be able to demonstrate what an atmosphere value is in our case, and to be able to show that isotope values are not drifting towards atmospheric ones in a systematic way.

L338: It would be nice to find the 500 ppmv threshold on the vertical axis of Figure 3. Also, I would prefer to have the horizontal axis start with "2".
We have added a red dashed horizontal line at the 500-ppm threshold on all three panels of Figure 3. We created a version of the figure where the axis starts with 2, but felt that the axis became too crowded to be readable. We have, however, submitted a wider version of the figure with the updated manuscript to make the figure more readable.

L346: These method details should appear in the method section, not in the results.
We moved the relevant information to the methods section.

L349: "we expect" sounds like a discussion statement (also in L362). What are the uncertainty limits and are they really transferable given the quite different setups (Oerter et al. vs. SWISS)?
We chose to keep this sentence in the manuscript because it is intended to help guide the reader to understand our plots. However, we have moved our discussion of why we think that

our system is within the uncertainty of the Oerter system into the discussion section. Ultimately, our system is very similar to the Oerter 2016 system in terms of how water vapor is produced and the plumbing, and so the idea is that the SWISS introduced no additional uncertainty.

L379f: This is already an interpretation of the results. Why is there an inter-unit variability? Or why do you expect one? And why is it (expected to be) different from the intra-unit variability? We have moved this to the discussion. The SWISS units are unique as they were made by hand, and so have small differences between the boxes (the ways we bent tubes, the flask placement inside the SWISS, greater experience with the tasks after building earlier SWISSs, etc.) that shouldn't have a large effect on the dataset, but seem to show up. We do a careful job of flagging flasks that might not be reliable, and we document the behavior of all the SWISS before they go into the field. This is why, in the discussion, we name the SWISS unit that got used for each experiment and the flasks that may be problematic, as well as general behaviors that might be an issue for a box.

Figure 4, 5, and 6: Please add the permil symbol to the vertical axes labels when presenting isotope data. Please add the 500 ppmv to the axis if this is the concentration threshold. Thank you for catching that. We have made those adjustments to figures 4, 5,6

L508: This is misleading as all flasks contain water vapor. Maybe you want to make the distinction between FLASH-EVAPORATED water vapor and atmospheric vapor? Thank you for this suggestion, we have adjusted the text to be "flash-evaporated water vapor" and "atmospheric vapor" both here and in the discussion.

L513 and elsewhere: I think table captions should be placed above the tables Thank you, we have made this adjustment

L533ff: Why were those data excluded? 13000 ppmv is well within the instrument's measurement range. If the soil had been very dry, aren't the obtained data then representative for this setting and can be interpreted accordingly? Do you mean atmosphere via leaks or by being present in the soil despite the huge gas-liquid interfacial area present even in quite dry soils and facilitating quasi-instant exchange between the liquid phase and a potentially intruding atmosphere?
This is a great question, and there are two key issues with these data. The first is a lab analysis issue - 13000 ppm starts to enter the range where there is a strong linearity effect in the measurement of stable isotope data. This linearity is challenging to correct for, especially when we use the dry-air carrier gas method where the water vapor mole fraction changes throughout the measurement. The second lies in uncertainty with how the vapor permeable tubing functions at very low soil moistures – some researchers have hypothesized that there is a greater injection of sampling gas ($N_2$) at low soil water contents, and so it is not totally clear how representative those isotope values are of a 'true' soil water isotope value. We have clarified our language and explanation in the results section.

L541f: How was condensation detected and why were these data excluded? Presenting (and later discussing) even spurious data in this rather technical manuscript would help future users of your setup to identify similar problems.
This is a great idea and agree that this would help future users identify similar issues. We have now included those data in both Figure 7 and table 3. We have also added discussion of those data in section 6.2.

L548: The discussion of the helium test is missing

There are no data to present from the helium testing per se, because the helium leak detector just gives on-the-spot indication that flasks have been broken. We will, however, include reference to how often we found broken flasks in the discussion as a way to address this comment and one other comment made on the supplemental material.

L551-556: Why is a seven-day test sufficient when the intended field-deployment is on a much longer timescale?

We think it is important that a seven-day dry air test is just one component of a longer QA/QC process. This concern is also why we completed 34 – 52-day dry air tests in a field setting. We have expanded in the manuscript on our philosophy of using the seven-day test as a reasonable compromise between effectively exposing leaks while being short enough duration that work can continue efficiently.

L557-561: How is it possible that not all of the atmospheric water vapor has been flushed given the achieved turnover rate? How can this be tested/identified? What solution would you suggest to prevent this from happening?

If there is atmospheric vapor in the lines during the measurement phase, it is possible to flush it out for 30 minutes prior to starting the measurement of flasks. We suspect that the effect actually is derived more from the 'filling' stage at the start of the experiment. of the Drierite was not pre-flushed well enough prior to the start of the experiments, and right after changing (refreshing) the Drierite, it is possible to have a large amount atmospheric vapor in the lines. In the manuscript we suggest that future users prevent this issue by flushing the lines with dry air longer, prior to both the start of the filling stage and of the measurement phase.
Regardless, the scale of the issue is quite small, given that all flasks still have water vapor mixing ratios of less than 700 ppm.

L564-570: This is a repetition from previous sections. Unfortunately, many paragraphs in the discussion start with a repetition of the results which should be avoided.

Thank you for this suggestion, we have removed the repetitious text.

L580: These data must be presented in the result section. And discussed here. I for one find it very remarkable that there is absolutely no variation in temperature with depth AND time for two of the three field sites. Any error in this variable would have an impact on the trustworthiness of vapor concentration and thus isotope readings, right? Also please add "(°C)" to the table heading of the temperature column.

Thank you so much for catching that error; we have re-input the correct temperature values which show expected variability. We have also added °C to the column header.
Error in this value would certainly have an impact on the isotope values through the vapor-liquid correction, though at this scale of variability the impact is small (~ 0.1‰ for $\delta^{18}O$ values and ~1‰ for $\delta^2H$ values) and well within the error of the overall system.

L586-601: This is not a proper discussion but mainly a repetition from previous sections.

We have significantly trimmed this text to remove unnecessary repetitious material. However, in this case we kept of some of the text to remind readers of the magnitude of the offset correction and uncertainty of the SWISS.

L599: What is the minimum acceptable concentration for this method? Given the overall analytical uncertainty, the additional uncertainty resulting from lower-than-optimum vapor concentrations should not make that much of a difference for quite a wide range, right?

Below ~17,000 ppm it would be important to consider a linearity correction, certainly below 13,000 ppm. For the dry-air carrier gas method it is tricky to apply a linearity correction because the water vapor mole fraction is decreasing throughout the measurement, and so the linearity correction would need to be applied 'continuously' to each data point taken every 1 hz. Moreover, the linearity correction is also highly non-linear across both composition and concentration, and so it becomes a 2-dimensional correction surface across which the data need to be corrected. Applying the 2-dimensional correction across every single 1-hz data point is non-trivial, and quantifying the relative contribution of the uncertainty is also non-trivial. We chose to not add additional text in this spot because of the emphasis on this point we have added elsewhere in the manuscript.

L607: are you referring to air or below-ground temperatures?

Thank you for catching that, we have clarified that it was air temperatures. We have also clarified that while the SWISS is stored below ground, it is not truly buried and so likely also experienced near freezing temperatures. Additionally, considering other comments, we have moved this text to the methods section.

L614: "quite" is too vague. Is the system sufficiently resistant to facilitate a specific uncertainty that allows for deciphering natural variations which – as you state elsewhere – become smaller with depth?

In this sentence we had been purposely vague, because we didn't want to over interpret these results. We do, however, understand your concern with the vagueness and have opted to take out that portion of the sentence.

L618: how was condensation noticed? This is valuable information for unexperienced users considering to employ your system.

We have now included a much longer explanation of this in our observations of condensation in section 6.2 of the discussion.

L620: "oxygen isotope" is redundant.

Thank you, fixed.

L624: Why is this surprising?

We have rephrased this to be "In contrast to the oxygen isotope results" to highlight that the two isotope systems don't seem to mimic each other, as might be expected.

L633: I think you didn't mention "replacing" in the method section. Only "tightening".

We have added in the methods section that sometimes replacing flasks helped as well as tightening.

L639: the observed "greater exchange and leaking" is based on the measurement of only two flasks (#10 is missing in the seven-day data, Figure SI 5) whereas three of the stainless steel fitted flasks (#3, #14, #15) reveal HIGHER vapor concentration than the PTFE fitted flasks after seven days and LOWER-than-before vapor concentrations after 27 days. How is the latter possible?

These were two sequential dry-air tests, and we have clarified this in the main text. We know that this is a very limited test, and we included it in the manuscript to be as transparent and helpful to other potential users as possible. We only present the PTFE fittings results as an example of something that may work, but that we didn't feel was worth pursuing.

L645: At what depth does the magnitude of seasonal variability have to be expected to fall below the system's analytical uncertainty? And what is the accuracy here?
It is soil dependent, but in many soils the signal may become too small below a depth of ~50 cm. So, the SWISS is still useful to answer many critical zone questions.

L648: In my understanding, vapor bound to the flask walls would introduce a memory effect which you excluded (L651). Did you really test for this by using flask with intentionally different filling histories for sampling identical vapor sources?
We removed the need for this experiment by cleaning and drying the flasks completely between each experiment.

L653-655: Why is the concentration benchmark of multiple dry air tests relevant and not the duration of the test itself?
We have clarified that we mean that one could use either longer tests or lower benchmarks that would indicate that flasks are less leaky.

L657: I'm struggling with "faithfully" because obviously there were non-tested factors causing the high failure rates during the actual application. Also, how would you identify outliers without an independent validation of soil water isotope data?
We have removed the word "faithfully". We have also added text in the discussion section 6.2 about identifying outliers. It can be difficult to identify outliers that arise from methodological errors or challenges when they aren't observed in progress, but multiple aspects of the broader datasets can help with this identification, and it is up to individual users to carefully design experiments so that confounding factors and outliers can be identified. For example, collecting environmental temperature, relative humidity and precipitation data can help shed light on whether data trends are reasonable given conditions. Sequential data also is useful because it helps identify samples that are unreasonably different from sequence neighbors given conditions.

L663: "Helpful"? Isn't it even mandatory in order to prevent condensation of the soil's vapor-saturated gas sample? Would a second Valco valve for the application of a constant zero-air dilution stream do the trick more reliably (similar to Volkmann & Weiler, doi: 10.5194/hess-18-1819-2014)? Or would this go beyond the budget?
The dilution stream in Volkmann and Weiler does more to control the water vapor mole fraction as it enters the Picarro, just as we did in the dry-air carrier sample introduction method, rather than to control condensation. It wouldn't prevent condensation in the sample line before the soil gas makes it to the SWISS.

L673: Please specify that the uncertainty refers to the hydrogen isotope values
Thank you, we have made that change

L682: All data should be presented in the results section
We have included the original discussion of these data in the results, and trimmed this section appropriately.

L686: How much is that in mm? Can you exclude convective precipitation with limited spatial extent or why was it "likely" (L687) indicative for precipitation at your field site?

On this part of the Nebraska plains storms frequently move northwest to southeast along the western edge of the Black Hills, moving over our field site towards the town of Chadron. We use the word 'likely', as opposed to 'highly likely' or 'certainly' here because commonly when it rains in Chadron it also rains at our field site, but not always.

L691: If any such precipitation event homogenizes the soil water isotope depth profile then how can you still assume only small seasonal variability at those depths? Wouldn't you then rather have to expect a significant shift in the soil water isotope depth profile after every precipitation event of similar magnitude? What other factors (soil moisture?, pore size distribution?) could be relevant in this context?

This is a really interesting comment, because it is going after some of the assumptions that something like the SWISS could be used to test. As a community, we have a limited number of datasets that show seasonal stable isotope variability at 50 cm across pore size distributions, soil moisture, clay mineralogies, etc. We don't have interannual datasets that can add dimensions of drought and rainfall timing variability, which may be an important factor for our datasets in the Western U.S. Rather than stating that homogenization occurs so concretely, we have rephrased this sentence to set up a testable hypothesis, and pointed to this as a need for future work.

L717: Results. Please move to and describe in the result section.

We have moved this figure to the results section, and appropriately described it in the results section.

L721f: Please also show discarded data in order to give readers of this rather technical paper the opportunity to learn from your experience. Then explain in detail what aspect(s) made you exclude these data from further consideration. What do you mean by "sufficiently"? According to Picarro's instrument specifications, even at 2500 ppmv the uncertainty induced by low vapor concentration is still below the SWISS overall uncertainty. If such low vapor concentrations were representative for very dry soil, wouldn't the SWISS still be able to deliver meaningful results?

We have included the discarded data from the Seibert site, because we agree that readers can learn from those data. We expanded our discussion of those data in greater detail in section 6.2. However, we have chosen to not show the discarded data from the Briggsdale site because it is a community standard to exclude data collected using the vapor permeable tubing from soil levels with volumetric water contents below 0.05, and because we were not able to appropriately correct for linearity affects during the measurement of those data.

L737: Can you provide details of this evidence? Which conditions caused condensation and how can it be avoided or otherwise dealt with?

We have added a paragraph describing these data in detail

L742: What does "CoAgMet" mean?

CoAgMet is the meteorological monitoring network that runs the meteorological instrumentation at the site, we have included greater detail on the site in the background section.

L748: The low d-excess value is not helpful in this context because it is not an independent measure but is calculated directly from the aforementioned oxygen and hydrogen isotope values.

We agree that the use of the word corroborated is confusing here, but we think that looking at these data from the perspective of d-excess rather than by value alone can still be helpful, so we have changed the wording of the sentence to indicate that more clearly.

L749: Again, please report such information in the results section.

We have moved this to the results.

L766: Why does uneven heating have an adverse effect? Wouldn't ANY heating be helpful as long as the induced temperature exceeds the sample's dew point, no matter by how much?

Uneven heating can lead to temperature gradients which might induce condensation at the point where the vapor meets slightly colder glass or stainless steel.

L768: Do you mean consistent amounts of heat or stable temperature?

We have clarified that we mean stable temperature in the manuscript.

L775: What problems would an IoT cellular router solve?

It would help us adjust the sampling plan on the fly so that we could avoid days where condensation might more readily form in the field or if there is a precipitation event that would be important to sample, or catch other technical problems that are worth a field visit to fix.

L778: Why is the inherent thermal structure of the soil important? How does it affect isotope measurements?

There are a few issues with heating the soil – first heating could potentially drive evaporation or diffusion in ways that are hard to account for, second it could potentially change the environment for plant and microbial life, which is of primary interest for many ecohydrology studies, or third, as in our case, variation in the temperature of the soils and how that is recorded by the mineralogy of the soil is a primary research question.

L783: "can be" sounds a little weak. Previously, you "strongly recommended" this procedure to be applied.

Thank you for this suggestion, we have made this adjustment.

Supplement: Please use similar formatting for the different figures.

We have ensured that the formatting for all of the figures matches.

Table SI 1: You emphasized the option to reproduce the SWISS setup (L169). Can you please list the monetary costs for different components or the entire SWISS setup as well?

We are happy to provide approximate costs in USD, but note that prices may vary

Table SI 2: Are you endorsing Amazon?

It was not our intention to endorse Amazon, rather it was our intention to communicate that there are many distributors of products that are all of equal quality. We have changed the text to read "Widely available product made by multiple manufacturers"

Table SI 3: Are the high standard deviation values for Toblerone after 52 days a result of noise or trends during measurements?

Those are the result of trends during the measurement – the leaky flasks have increasing water vapor concentration through the measurement.

Supplement section 1: Build-out description: How many flasks were ultimately broken before the desired setup was achieved?

We typically broke 1-2 flasks during the building of SWISSs, and we've broken 2 flasks in transit to the field over ~8 deployments.

---

## Author Response (AR2)

The authors thank the reviewer for these comments that have improved the manuscript. We've have responded to each line by line comment below.

L153: please define the abbreviation "CRDS" upon first appearance. (You did so in a text segment that has been deleted.)

    We've added that text.

L388: I suppose you want to swap "suspicious" and "those". Or delete "those".

    We have removed the word "those". Thank you for catching that.

L439: VSMOW-SLAP isotope scale instead of VSMOW isotope scale

    We have made that adjustment, thank you.

L588: "consistent with evaporative enrichment" - I this an observation, interpretation, or a general statement? Please specify. ("typically consistent"?)

    We have added that it is a general statement.

L615: Please indicate here how "condensation" was observed/identified.

    We opted to take out this sentence, because we deal with the issue of condensation for thoroughly in the discussion section.

L668: I assume "IsoWagon" refers to the Oerter system? Please specify.

    It is, yes, we have added reference to that paper.

L802/3: move parentheses to after "dual isotopes"

    We have moved the parentheses, thank you.

L808: Is "vapor bound to the flask walls" an observation or a hypothesis? Please specify.

    This is a hypothesis, and have added reference to that in line 807.

L868: "than THE ONES FROM the other two sampling depths"

    We have added that text.

L903: Wouldn't condensation shift isotope data to the lower left since heavy isotope have preferentially left the vapor phase that is left for analysis?

    As stated this was a little confusing, so we've added a clarifying clause that helps to explain this mechanism. "It is possible that environmental conditions encouraged the formation of condensation in the impermeable tubing at the time of sampling;. if there was residual condensation in the impermeable tubing then its possible we were partially sampling a heavier condensed water."

L926: "a slightly colder spot" or rather "a spot slightly colder than its dew point"?

    Colder than dew point is more precise, so we have added that language

L933: Preventing condensation would not only reduce uncertainty but also the number of samples that need to be discarded.

Great point, we've added that text.

L941: remove the second "form"

We've removed that second "form"

L949: shed MORE light

We've added that text.